# CONTINUOUS AUDIO LANGUAGE MODELS

**Simon Rouard**[*]
Kyutai
UMR STMS
IRCAM-CNRS Sorbonne Univ.
simon@kyutai.org

**Manu Orsini**[*]
Kyutai
manu@kyutai.org

**Axel Roebel**
UMR STMS
IRCAM-CNRS Sorbonne Univ.

**Neil Zeghidour**
Kyutai

**Alexandre Défossez**
Kyutai
alex@kyutai.org

## ABSTRACT

Audio Language Models (ALM) have emerged as the dominant paradigm for speech and music generation by representing audio as sequences of discrete tokens. Yet, unlike text tokens, which are invertible, audio tokens are extracted from lossy codecs with a limited bitrate. As a consequence, increasing audio quality requires generating more tokens, which imposes a trade-off between fidelity and computational cost. We address this issue by studying Continuous Audio Language Models (CALM). These models instantiate a large Transformer backbone that produces a contextual embedding at every timestep. This sequential information then conditions an MLP that generates the next continuous frame of an audio VAE through consistency modeling. By avoiding lossy compression, CALM achieves higher quality at lower computational cost than their discrete counterpart. Experiments on speech and music demonstrate improved efficiency and fidelity over state-of-the-art discrete audio language models, facilitating lightweight, high-quality audio generation. Samples are available at iclr-continuous-audio-language-models.github.io. Finally, we release Pocket TTS, an open-source 100M-parameter text-to-speech model that can run faster than real time on a laptop CPU: github.com/kyutai-labs/pocket-tts

## 1 INTRODUCTION

Using classification over a finite vocabulary as the training objective for autoregressive sequence models is an effective approach for naturally discrete modalities such as text, where large-scale Transformer-based (Vaswani et al., 2017) language models such as LLaMa (Touvron et al., 2023) and GPT-4 OpenAI (2024) have achieved impressive results. To extend this powerful framework to continuous domains such as image, audio, or video, previous work has mostly relied on discretizing signals using lossy compression algorithms (van den Oord et al., 2018), such that they become akin to text. In particular, neural audio codecs (Zeghidour et al., 2021; Défossez et al., 2024a) have provided discrete representations of audio that are compact enough to allow for high-quality speech (Borsos et al., 2023; Wang et al., 2023) and music (Agostinelli et al., 2023; Copet et al., 2023) generation with autoregressive models. In this context, a Residual Vector Quantizer (RVQ) (Zeghidour et al., 2021) transforms an audio frame into a coarse-to-fine hierarchy of tokens. As quantization inevitably introduces a perceptual quality loss, generating high-fidelity audio requires increasing the bitrate of audio tokens, which amounts to using deeper hierarchies of RVQ tokens. A consequence of growing the size of the token matrix (along time and token depth) is an additional computational load for the generative model, as the strong dependencies between tokens of the same frame(Lemercier et al., 2024) prevent fully parallel generation. The naive approach of flattening the token hierarchy (Borsos et al., 2023) being prohibitively expensive, Copet et al. (2023) introduces a delay pattern that conjugates the computational efficiency of parallel generation with a better modeling of inter-token dependencies. Lee et al. (2022) and Yang et al. (2023) furthermore introduce a smaller RQ-Transformer model that is

---

[*]Equal Contribution

autoregressive along the depth axis, and Défossez et al. (2024b) combines this approach with the delay pattern. While these methods currently power state-of-the-art generative models for audio (Défossez et al., 2024b; Labiausse et al., 2025), the trade-off imposed by residual quantization between quality and computation remains too constraining for generating high-quality audio on edge devices.

This motivates an alternative strategy: autoregressive modeling of continuous latents without quantization. Standard variational autoencoders (VAEs) are easier to train, are not affected by issues such as codebook collapse, and can reconstruct audio at higher fidelity for the same latent dimensionality. Pioneering work in the vision domain that autoregressively models continuous sequences includes GIVT (Tschannen et al., 2024) and MAR (Li et al., 2024), followed by some larger models in the image domain (Fan et al., 2025; Gu et al., 2025) and attempts in the audio domain (Turetzky et al., 2024; Pasini et al., 2024b; Jia et al., 2025; wen Yang et al., 2025). In MAR, the authors model the per-token probability distribution with a diffusion model (in the form of a small MLP) conditioned on a latent variable modeled by an autoregressive transformer backbone. SALAD (Turetzky et al., 2024) and DiTAR (Jia et al., 2025) adapt MAR-style diffusion heads for text-to-speech modeling, achieving better audio quality than discrete baselines. However, these works are limited to text-to-speech on small-scale and domain-specific datasets, leaving open the question of how well they can adapt to more complex tasks such as speech continuation (without text supervision) and richer audio domains such as music. We apply CALM to 4 tasks which are speech and music continuation as well as text-to-speech and text-to-music.

We propose Continuous Audio Language Models (CALM) that predict sequences in the latent space of a VAE, bypassing the need for quantization. While we build on the MAR architecture where a transformer backbone uses $(\mathbf{x}^1, \ldots, \mathbf{x}^{s-1})$ to predict an intermediate latent $\mathbf{z}^s$, which then conditions a head (diffusion model) that models $p(\mathbf{x}^s|\mathbf{z}^s)$, we find that without further improvements, it fails to generate rich audio content and is slow to sample from. To overcome this, we introduce several key innovations:

**1. Improving quality and stability**: To mitigate error accumulation during inference, we follow Pasini et al. (2024b) and, during training, inject noise into the long-term context $(\mathbf{x}^1, \ldots, \mathbf{x}^{s-1})$. Additionally, we introduce a short-context transformer that summarizes recent clean latents, providing the sampling head with both coarse long-range context and fine-grained local information. **2. Diffusion-to-Consistency replacement**: We replace the diffusion model with a continuous consistency model (Lu & Song, 2025) during training, significantly accelerating inference without compromising sample quality. This change reduces the inference time of the sampler head by a factor of up to $\times 20$ in our music experiments and $\times 12$ in our speech experiments compared to an RQ-Transformer head. **3. Gaussian Temperature sampling**: Temperature control is crucial for high-quality speech generation, yet consistency models lack a formal mechanism for temperature sampling. We present a heuristic that approximates temperature sampling in the consistency setup. **4. Head batch multiplier**: Sampling multiple noise levels at training time for the same latent highly accelerates training for a small cost. **5. Latent Classifier Free Guidance**: we apply Classifier Free Guidance to the latent variable conditioning the consistency head for the conditioned CALM. **6. Latent Distillation**: once that we have chosen a latent CFG coefficient, we can distill the CFG computation of the backbone into a student backbone while keeping the same sampling head (MLP), hence dividing the batch size by 2 at inference time. This distillation can also be applied to a much smaller student backbone.

Finally, by using these innovations, we introduce **Pocket TTS** which is a 100M-parameters text-to-speech model that can run faster than real-time on a laptop CPU. We detail the results of **Pocket TTS** in Sec.G and in the following technical report: kyutai.org/pocket-tts-technical-report.

## 2 RELATED WORK

**Autoregressive audio language models.** Early autoregressive audio models operated on raw waveforms such as WaveNet (van den Oord et al., 2016), learned discrete codes via VQ-VAE, such as in Jukebox (Dhariwal et al., 2020) or continuous word-sized audio tokens (Algayres et al., 2023). The advent of neural audio codecs (Zeghidour et al., 2021; Défossez et al., 2024a) enabled high-quality vector-quantized tokenizers for general audio. These, in turn, powered discrete-token audio LMs: AudioLM (Borsos et al., 2023) for unconditioned audio generation while AudioGen (Kreuk et al., 2023), MusicLM (Agostinelli et al., 2023), and MusicGen (Copet et al., 2023) apply similar methods for text-to-audio and text-to-music generation. In speech, autoregressive modeling of RVQ codes

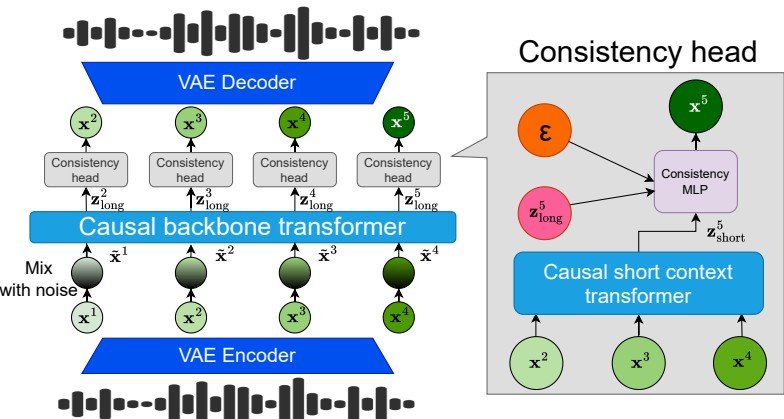

Figure 1: Overview of our model. During training, latent vectors $\mathbf{x}^s$ are noised to encourage the backbone Transformer to focus on coarse structure. The consistency head is a consistency model conditioned on the latent variable $\mathbf{z}^{\mathbf{s}}_{\text{long}}$ produced by the backbone, as well as a short-term context vector $\mathbf{z}^{\mathbf{s}}_{\text{short}}$ computed from a short-context Transformer applied to the most recent clean latent tokens.

has been used for text-to-speech generation (Wang et al., 2023; Kharitonov et al., 2023) as well as for spoken dialogue (Défossez et al., 2024b) or translation (Labiausse et al., 2025). However, all of these systems rely on lossy quantization, which inevitably degrades audio fidelity unless a significant compute budget is spent to generate a deep hierarchy of RVQ tokens. This is unlike CALM, which predicts continuous embeddings in one pass, providing a better quality-computation trade-off.

**Continuous autoregressive models.** In GIVT, Tschannen et al. (2024) use a transformer to autoregressively model the latent space of a VAE trained on images, parameterize a Gaussian Mixture Model, and train it with a cross-entropy loss. The authors of MAR (Li et al., 2024) obtain better results by replacing the Gaussian mixture model with a diffusion-model enabling the approximation of more diverse distributions. In MAR, a large transformer backbone predicts a continuous embedding $\mathbf{z}^s$ given $(\mathbf{x}^1, \ldots, \mathbf{x}^{s-1})$, which then conditions a small MLP diffusion network that models the probability distribution $p(\mathbf{x}^s | \mathbf{z}^s)$ of the next latent. This eliminates the need for discrete tokenizers, but at the cost of slow sampling: MAR typically needs hundreds of denoising steps per token. Hang et al. (2025) aims to speed up MAR by replacing the diffusion head with a shortcut head (Frans et al., 2025) for few-step sampling. Shortcut models combine a diffusion loss and a self-consistency loss in order to accelerate the diffusion process. They reduce the number of diffusion steps from 100 to 8 with a similar image quality. Remarkably, CALM achieves a quality comparable to the best discrete models with only one step of consistency modeling.

In audio, the approach of MAR has been adapted for the task of Text-to-Speech (TTS) synthesis. SALAD (Turetzky et al., 2024) introduces a zero-shot TTS model that operates on continuous speech representations using a per-token latent diffusion process. By leveraging semantic tokens for contextual information and determining synthesis stopping points, SALAD achieves improved intelligibility and audio quality without relying on quantization. Similarly, DiTAR (Jia et al., 2025) presents a patch-based autoregressive framework combining a language model with a diffusion transformer for speech generation. This approach models dependencies between aggregated local patches of continuous tokens, using a causal language model to produce embeddings, which, along with previous patches, serve as inputs to a bidirectional diffusion transformer that predicts the next patch. The authors observe that providing local context through patching was determinant to improve their model, which corroborates what we observe with the introduction of a short context transformer into our model. In Pasini et al. (2024b), the authors apply the MAR framework to music generation using a relatively small dataset comprising 20,000 single-instrument stems, training their model on 10-second excerpts on top of their continuous compression model Music2Latent (Pasini et al., 2024a). They introduce a method for noise augmentation of the data that allows the model to avoid error accumulation. However, we notice that scaling to a more complex and diverse dataset consisting of full musical pieces makes their approach struggle to maintain high-quality generation on longer sequences. To address these challenges, we propose novel strategies that enhance generation quality while also improving in-

ference efficiency. Finally, Music2Latent2 (Pasini et al., 2025) explores combining autoregressive and consistency modeling but for compression. IMPACT (Huang et al., 2025) explores MAR decoding for text-to-audio generation and sets a new standard for short latency models on the AudioCaps (Kim et al., 2019) benchmark. More recently, MingUni-Audio (Yan et al., 2025) shows that continuous speech language models can scale to 20B Mixture of Experts models with 3B active parameters.

Some other autoregressive speech models work in the continuous domain thanks to spectral representations such as MELLE (Meng et al., 2024) for TTS and Flow-Omni (Yuan et al., 2024) for speech-to-speech conversation.

## 3 Background

**Notations:** Let $W \in \mathbb{R}^{f_s \cdot d}$ be a monophonic waveform of $d$ seconds sampled at frame rate $f_s$. Our goal is to model $W$ either in the discrete latent space of a RVQ-based codec or in the latent space of a VAE. Let $f_r$ be the frame rate of the codec or VAE.

In the case of the **discrete modeling**, $W$ is represented by the sequence of discrete tokens $\left(q^{s,k}\right)$, where $s \in \{1, \ldots, S\}$ indexes time and $k \in \{1, \ldots, K\}$ indexes codebook depth and $S = f_r \cdot d$. Each token $q^{s,k} \in \{1, \ldots, N_k\}$ is drawn from a finite vocabulary.

In the case of **continuous modeling**, $W$ is represented by a sequence $\left(\mathbf{x}^1, \ldots, \mathbf{x}^S\right)$ with $S = f_r \cdot d$ and $\mathbf{x}^s \in \mathbb{R}^C$ where $C$ is the latent dimension of the VAE.

### 3.1 Autoregressive Modeling with Residual Vector Quantization based codecs.

Autoregressive modeling of discrete tokens from RVQ-based codecs (Zeghidour et al., 2021; Défossez et al., 2024a) is a prevalent method for high-fidelity audio generation (Copet et al., 2023; Agostinelli et al., 2023; Borsos et al., 2023; Kreuk et al., 2023). Given a sequence of discrete tokens $\left(q^{s,k}\right)_{i \in \{1, \ldots, N\}, k \in \{1, \ldots, K\}}$, early models like Borsos et al. (2023); Agostinelli et al. (2023) flattened the multi-level sequence, increasing its length by a factor of $K$ and resulting in high computational costs due to the quadratic complexity of Transformer self-attention. MusicGen (Copet et al., 2023) mitigates this by using a delay pattern to independently sample each of the $K$ RVQ levels, adding only $K - 1$ tokens to the sequence but introducing a fixed latency of $K - 1$ frames, which is problematic for real-time applications. RQ-Transformer (Lee et al., 2022) addresses this by using a sampler transformer head that models the RVQ at a given time step, enabling low-latency generation.

Denoting $\mathbf{q}^s = (q^{s,1}, \ldots, q^{s,K})$, the *Backbone Transformer* $T_\theta$ encodes the history of previous timesteps to produce a context vector $\mathbf{z}^s$, and a *RQ-Transformer* $g_\phi$ autoregressively decodes the residual-wise components of the token stack at each time step. The context vector is then given by $\mathbf{z}^s = T_\theta(\mathbf{q}^1, \ldots, \mathbf{q}^{s-1})$, and the logits $\ell^{s,k}$ for predicting the $k$-th codebook token are computed as $\ell^{s,1} = \text{Lin}(\mathbf{z}^s)$ and $\ell^{s,k} = g_\phi\left(\mathbf{z}^s, q^{s,1}, \ldots, q^{s,k-1}\right)$ for $k > 1$.

These logits are trained using a cross-entropy loss over discrete tokens $\mathcal{L}_{\text{CE}} = -\sum_{s=1}^{S} \sum_{k=1}^{K} \log p\left(q^{s,k} \mid \mathbf{q}^{<s}, \mathbf{q}^{s,<k}\right)$.

This approach enables efficient parallel modeling of RVQ sequences, allowing all codebooks corresponding to a single timestep to be generated simultaneously without introducing any delay. However, a key limitation lies in the use of the *RQ-Transformer*, which is computationally intensive; as its resource requirements scale with the number of RVQ, or even with the square of the number of RVQ, if the attention dominates the computation cost.

### 3.2 Consistency Models

**Flow Matching and Probability Flow ODE.** Let $p_{\text{data}}$ be a data distribution over $\mathbb{R}^d$. Given $\mathbf{x}_0 \sim p_{\text{data}}$, diffusion (Ho et al., 2020; Song et al., 2021) and flow matching (Lipman et al., 2023) models define a forward noising process that gradually perturbs samples from $p_{\text{data}}$ through the noising process $\mathbf{x}_t = \alpha_t \mathbf{x}_0 + \sigma_t \boldsymbol{\epsilon}$, where $\boldsymbol{\epsilon} \sim \mathcal{N}(0, \mathbf{I})$ and $t \in [0, T]$. $\alpha_t, \sigma_t$ are predefined functions such that $\alpha$ is decreasing with $\alpha_0 = 1, \alpha_T = 0$ and $\sigma$ is increasing with

$\sigma_0 = 0, \sigma_T = 1$. In Flow Matching, a neural network $F_\phi$ is trained to minimize the loss $\mathcal{L}_{\text{FM}}(\phi) = \mathbb{E}_{\mathbf{x}_0 \sim p_{\text{data}}, \, \boldsymbol{\epsilon} \sim \mathcal{N}(0,\mathbf{I}), \, t \sim \mathcal{U}(0,1)} \left[ w(t) \left\| F_\phi(\mathbf{x}_t, t) - (\alpha_t' \mathbf{x}_0 + \sigma_t' \boldsymbol{\epsilon}) \right\|_2^2 \right]$.

Once trained, sample generation is performed by solving a deterministic ordinary differential equation known as the probability flow ODE (PF-ODE), which defines a continuous path from noise to data. In the context of Flow Matching, the PF-ODE is $\frac{d\mathbf{x}_t}{dt} = F_\phi(\mathbf{x}_t, t)$ with $\mathbf{x}_T \sim \mathcal{N}(0, \mathbf{I})$.

A **Continuous-Time Consistency Models.** (Song et al., 2023) is a neural network $f_\phi(\mathbf{x}_t, t)$ trained to map a noisy input $\mathbf{x}_t$ directly to the corresponding clean data $\mathbf{x}_0$ in a single step, by approximating the sampling trajectory of the probability flow ODE (PF-ODE) starting from $\mathbf{x}_t$. To ensure correct behavior, $f_\phi$ must satisfy the boundary condition $f_\phi(\mathbf{x}, 0) = \mathbf{x}$, thus leading to the common parameterization $f_\phi(\mathbf{x}_t, t) = c_{\text{skip}}(t)\mathbf{x}_t + c_{\text{out}}(t)F_\phi(\mathbf{x}_t, t)$, where $F_\phi$ is a neural network and the coefficients satisfy $c_{\text{skip}}(0) = 1$ and $c_{\text{out}}(0) = 0$ to fulfill the boundary condition.

By using $T = \frac{\pi}{2}$ and $\alpha_t = \cos(t), \sigma_t = \sin(t)$, Lu & Song (2025) derive the following continuous-time consistency loss where $w_\psi(t)$ is an adaptive weighting function:

$$\mathcal{L}_{\text{CM}}(\phi, \psi) = \mathbb{E}_{\mathbf{x}_t, t} \left[ \frac{e^{w_\psi(t)}}{D} \left\| F_\phi(\mathbf{x}_t, t) - F_{\phi^-}(\mathbf{x}_t, t) - \cos(t)\frac{df_{\phi^-}(\mathbf{x}_t, t)}{dt} \right\|_2^2 - w_\psi(t) \right]. \quad (1)$$

**Lagrangian Self-Distillation.** We also explore a new 1-step flow-matching method named Latent Self-Distillation (LSD) that has been introduced in Boffi et al. (2025). In this paper, the authors unify most 1-step methods into 2 categories and develops a third one (LSD) which appears to be more stable at training. See Sec. A for the equations.

### 3.3 Autoregressive Modeling of Continuous Latents via Diffusion.

Li et al. (2024) propose MAR, a method for autoregressive modeling over a sequence $(\mathbf{x}^1, \dots, \mathbf{x}^S)$ of continuous latent vectors extracted from a pretrained VAE, thereby eliminating the need for discrete quantization. As in the discrete case, a *Backbone Transformer* $T_\theta$ maps the context to an embedding: $\mathbf{z}^s = T_\theta(\mathbf{x}^1, \dots, \mathbf{x}^{s-1})$.

Then, a diffusion process parameterized by a neural network $\boldsymbol{\epsilon}_\phi$ is trained on each $\mathbf{x}^s$ with the loss $\mathcal{L}_{\text{diff}}(\theta, \phi) = \sum_{s=1}^S \mathbb{E}_{\boldsymbol{\epsilon} \sim \mathcal{N}(0,I), t \sim [0,1]} \left[ \|\boldsymbol{\epsilon} - \boldsymbol{\epsilon}_\phi(\mathbf{x}_t^s, \mathbf{z}^s, t)\|^2 \right]$ where $\mathbf{x}_t^s$ is a noisy version of $\mathbf{x}^s$ at diffusion timestep $t$: $\mathbf{x}_t^s = \alpha_t \mathbf{x}^s + \sigma_t \boldsymbol{\epsilon}$ with $\boldsymbol{\epsilon} \sim \mathcal{N}(0, I)$ and $\alpha_t$ and $\sigma_t$ are predefined schedules for all $t \in [0, 1]$. In practice, an MLP significantly smaller than the backbone transformer estimates $\boldsymbol{\epsilon}_\phi$. This replaces the categorical prediction used in discrete models (done by the RQ-Transformer) with a denoising task in the continuous domain (done by the MLP). This method enables flexible and differentiable modeling of continuous signals without requiring to perform quantization on the latent space which can lead to several issues such as codebook collapse, balancing quantization losses and training instabilities. A key limitation of this approach is that sample quality depends on the number of diffusion steps at inference, raising the question of whether it can surpass the RQ-Transformer under similar computational constraints.

## 4 Method

### 4.1 Our VAE-GAN

Most autoregressive audio models are built upon RVQ-GAN architectures (Zeghidour et al., 2021; Défossez et al., 2024a; Kumar et al., 2023; Guo et al., 2025). Following the approach of Evans et al. (2024), we instead adopt a VAE-GAN framework, replacing the RVQ bottleneck with a VAE bottleneck to regularize the latent space and enforce a Gaussian prior. Our VAE is fully causal and draws from the architecture of Mimi (Défossez et al., 2024b), using Transformers in addition to convolutions in the encoder and decoder, which have been shown to improve performance.

While training the model with adversarial losses and VAE regularization without any reconstruction losses improves the quality of the model for speech, it degrades the reconstruction quality for music. Semantic distillation is performed for the speech VAE similarly as in Mimi, with WavLM (Chen

et al., 2021b) as teacher. There is no semantic distillation for the music model and we let this for future work as semantic content is harder to define for music. The loss is:

$$\mathcal{L}_{\text{VAE}} = \lambda_{\text{t}}\mathcal{L}_{\text{t}}(x, \hat{x}) + \lambda_{\text{f}}\mathcal{L}_{\text{f}}(x, \hat{x}) + \lambda_{\text{adv}}\mathcal{L}_{\text{adv}}(\hat{x}) + \lambda_{\text{feat}}\mathcal{L}_{\text{feat}}(x, \hat{x}) + \lambda_{\text{KL}}\mathcal{L}_{\text{KL}} + \lambda_{\text{distill}}\mathcal{L}_{\text{distill}} \quad (2)$$

where $\mathcal{L}_{\text{t}}$ and $\mathcal{L}_{\text{f}}$ are the temporal and frequential reconstruction losses, $\mathcal{L}_{\text{adv}}$ is the adversarial loss, $\mathcal{L}_{\text{feat}}$ is the feature matching loss, $\mathcal{L}_{\text{KL}}$ is the KL regularization applied to the VAE bottleneck, and $\mathcal{L}_{\text{distill}}$ is the WavLM distillation loss applied for the speech VAE.

## 4.2 Continuous Audio Language Model (CALM) architecture

Let $\left(\mathbf{x}^1, \ldots, \mathbf{x}^S\right)$ denote the sequence of continuous latent vectors produced by a VAE encoder. As illustrated in Fig. 1, our model comprises three main components. The motivations behind these design choices are described in Sec. 4.3 and the ablation study in Tab. 6 justifies these choices.

**1. Causal Backbone Transformer with Noise Injection**  We build on the MAR framework (Li et al., 2024) by employing a causal Transformer $T_{\text{long},\theta^1}$ to capture long-term dependencies. However, during preliminary experiments, we realized that music generation models with the MAR framework were generating poor quality audio and diverging quickly at inference because they were not robust to error accumulation. Pasini et al. (2024b) introduced a noise augmentation trick at training time in order to tackle this problem. Given a sequence $(\mathbf{x}^1, \mathbf{x}^2, \ldots, \mathbf{x}^S)$, they sample $k_s \sim \mathcal{U}(0, 1)$ and $\epsilon_s \sim \mathcal{N}(0, \mathbf{I})$ for every $s \in \{1, \ldots, S\}$ and use a noised input to the backbone $\tilde{\mathbf{x}}^s = k_s\epsilon_s + (1-k_s)\mathbf{x}^s$ for every $s$. Early experiments showed that preserving the variance of $\tilde{\mathbf{x}}^s$ improved quality, so that we use instead $\tilde{\mathbf{x}}^s = \sqrt{k_s}\epsilon_s + \sqrt{1 - k_s}\mathbf{x}^s$. We don't perform any noise augmentation at inference time. Thus, $\mathbf{z}^s_{\text{long}} = T_{\text{long},\theta^1}\left(\tilde{\mathbf{x}}^1, \ldots, \tilde{\mathbf{x}}^{s-1}\right)$. Noise injection prevents error accumulation during inference, but as shown in Tab. 6 is insufficient alone for high-quality music generation.

**2. Short-Context Transformer**  To supply local, high-resolution context to the denoising head, we introduce a lightweight causal Transformer that attends the $K$ previous clean latents (we use $K = 10, \sim 0.4s$ of music): $\mathbf{z}^s_{\text{short}} = T_{\text{short},\theta^2}\left(\mathbf{x}^{s-K}, \ldots, \mathbf{x}^{s-1}\right)$. This short-context embedding $\mathbf{z}^s_{\text{short}}$ supplies fine-grained information potentially lost through noise injection in the backbone. We show in Sec. E.3 that the value of K is not a decisive hyperparameter but Tab. 6 indicates that the short-context transformer is crucial to good quality generation.

**3. Consistency-Model Head**  Finally, a small MLP-based consistency model $f_\phi$ is conditioned on the sum of the long-term and short-term features, $\mathbf{Z}^s = \mathbf{z}^s_{\text{long}} + \mathbf{z}^s_{\text{short}}$. At inference time, for 1-step generation, the next latent $\hat{\mathbf{x}}^s$ is sampled through: $\epsilon \sim \mathcal{N}(0, I), t = 1, \hat{\mathbf{x}}^s = f_\phi\left(\mathbf{x}^s_1 = \epsilon, t = 1, \mathbf{Z}^s\right)$.

In addition to consistency, we experiment with the TrigFlow (Lu & Song, 2025) formulation of flow-matching for the MLP. Although TrigFlow yields marginally higher fidelity, its inference cost makes it impractical for real-time use. While Tab. 4 shows it, this tradeoff is studied in Sec. E.4.

Together, these three components form a continuous autoregressive model that (i) leverages noise-robust long-term modeling, (ii) preserves local detail via short-context conditioning, and (iii) achieves rapid, high-fidelity latent sampling through consistency modeling.

The training objective for one sequence $\left(\mathbf{x}^1, \ldots, \mathbf{x}^S\right)$ is defined by:

$$\mathcal{L}_{\text{CALM}}(\theta, \phi, \psi) = \sum_{s=1}^{S}\mathbb{E}_{t,\epsilon}\left[\frac{e^{w_\psi(t)}}{D}\left\|F_\phi\left(\mathbf{x}^s_t, t, \mathbf{Z}^s\right) - F_{\bar{\phi}}\left(\mathbf{x}^s_t, t, \mathbf{Z}^s\right) - \cos(t)\frac{df_{\bar{\phi}}(\mathbf{x}^s_t, t, \mathbf{Z}^s)}{dt}\right\|^2_2 - w_\psi(t)\right],$$

$$(3)$$

where $\mathbf{Z}^s = \mathbf{z}^s_{\text{long}} + \mathbf{z}^s_{\text{short}} = T_{\text{long},\theta^1}\left(\tilde{\mathbf{x}}^1, \ldots, \tilde{\mathbf{x}}^{s-1}\right) + T_{\text{short},\theta^2}\left(\mathbf{x}^{s-K}, \ldots, \mathbf{x}^{s-1}\right)$,
$t \sim [0, 1], \epsilon \sim \mathcal{N}(0, I)$ and $\mathbf{x}^s_t = \cos(t)\mathbf{x}^s + \sin(t)\epsilon$. All the parameters $(\theta, \phi, \psi)$ of the transformer backbone $T_{\text{long},\theta^1}$, the short-context transformer $T_{\text{short},\theta^2}$, the consistency MLP $f_\phi$ and the adaptive weighting function $w_\psi$ are jointly trained together with this consistency loss similarly as the backbone and the RQ-Transformer are trained through cross-entropy loss in the discrete case.

### 4.3 Combining noisy long-term context and clean short-term context

In preliminary experiments, music generation models trained with the MAR framework were diverging quickly during inference because they were not robust to error accumulations. Applying noise injection during training slightly improves model stability but often reduces detail and instrument diversity, typically preserving only rhythmic elements, with audio fading into silence after 10–15 seconds. Since Pasini et al. (2024b) targets short, single-instrument clips, it's unsurprising the method performs best in that constrained setting. We hypothesize that the added noise inhibits the backbone transformer from encoding fine-grained information into $\mathbf{z}_{\text{long}}^s$, limiting the MLP's ability to reconstruct detailed audio. However, combining this with a short-context transformer computing $\mathbf{z}_{\text{short}}^s$ yields the best results (Tab. 6), likely because the clean short-term context restores local detail needed to model the distribution of the next $\mathbf{x}^s$.

### 4.4 Head Batch Multiplier

Training is bottlenecked by the cost of generating the conditioning variable $\mathbf{z}_{\text{long}}^s$ via the large causal transformer. To address this, we introduce the *Head Batch Multiplier*, which amortizes this cost by reusing $\mathbf{z}_{\text{long}}^s$ multiple times per training step. Specifically, for each input sequence, we compute $\mathbf{z}_{\text{long}}^s$ once and use it across $N$ loss computations, each with independently sampled noise levels $t$ and $\epsilon$. This improves efficiency and stabilizes training by averaging the loss over multiple samples. Tab. 6 and Fig. 3, show faster convergence and better final performance at comparable training cost.

### 4.5 Gaussian temperature sampling

Sampling strategies, such as temperature sampling, have a significant impact on generation quality in the discrete setup, particularly for speech. To replicate this behavior in the continuous domain, we introduce a sampling heuristic that results in comparable gains. Similarly to the GAN noise truncation trick presented in Brock et al. (2018), we sample more from the high probability zone of the Gaussian to trade diversity for fidelity.

While the GAN truncation trick truncates the Gaussian noise such that values outside of a certain range are redrawn, we chose to reduce the variance of the Gaussian noise instead. This is mathematically equivalent to applying a temperature $\tau$ to the Gaussian if we change the standard deviation to $\sqrt{\tau}$. This makes temperature values between the discrete and continuous setups somewhat comparable, and we found that using a temperature of .8 for speech continuation was bringing good results in both setups. The effects of gaussian temperature are further discussed in Section C.

### 4.6 Latent Classifier Free Guidance

Classifier Free Guidance (CFG) (Ho & Salimans, 2022) is known to improve the generation quality of conditioned generative models. It can be applied for diffusion and flow matching models on the sampling trajectory as well as on the logits of autoregressive language models (Kreuk et al., 2023). Since CFG cannot be applied on the trajectory of 1-step consistency models we decide to apply the CFG on the outputs of the Backbone and Short-Context transformers. Formally, given $\mathbf{C}$ a conditioning and $\alpha$ the CFG coefficient, we compute for every $s$ of the sequence $\mathbf{Z}_{\text{CFG}}^s = \mathbf{Z}_\emptyset^s + \alpha(\mathbf{Z}_C^s - \mathbf{Z}_\emptyset^s)$ and then we generate $\hat{\mathbf{x}}^s$ with the consistency head conditioned on $\mathbf{Z}_{\text{CFG}}^s$. We call this method Latent CFG, as it operates on the latent variable $\mathbf{Z}^s$ instead of the model output. It has been introduced in the video-to-audio model SoundReactor (Saito et al., 2025).

### 4.7 Latent Distillation

Once a teacher model has been trained and the desired classifier free guidance (CFG) coefficient has been selected for inference, we distill the CFG-guided teacher into a student model to avoid the need to double the batch size during inference when using CFG. To this end, we distill only the backbone transformer and directly copy the teacher's MLP head into the student.

The distillation objective for the student backbone is an $\ell_2$ loss between the latent representation $\mathbf{Z}_{\text{distill}}^s$ produced by the student backbone and the CFG-guided latent representation $\mathbf{Z}_{\text{CFG}}^s$ produced by the teacher. Additionally, the student backbone transformer may contain fewer layers than the teacher.

In practice for Pocket TTS, we distill a text-to-speech model with 24 transformer layers into a student model with only 6 layers, using a latent CFG coefficient of $\alpha = 1.5$. See Sec. G for more details.

## 5 EXPERIMENTS AND RESULTS

### 5.1 SPEECH CONTINUATION

**VAE:** Our VAE is based on Mimi (Défossez et al., 2024b) but enforces gaussian inner latents instead of a categorical distribution. Like in Mimi, to enforce semanticity of the representations, we distill WavLM into the inner latent representation with a cosine similarity loss. Unlike Mimi, which applies this loss only to the first codebook, we extend it to the entire latent representation.

Table 1: **Speech compression models.** Our VAE is on par with the VQ-VAE on acoustic quality (MOSNeT) and outperforms it on semantic discriminability (ABX), PESQ Rix et al. (2001) and STOI Taal et al. (2011) and a MUSHRA for acoustic quality.

| MODEL TYPE | DIMS / RVQ | FRAME RATE (HZ) | BITRATE (KBIT/S) | MOSNET (↑) | ABX (↓) | PESQ (↑) | STOI (↑) | ACOUSTIC QUAL. (↑) |
|---|---|---|---|---|---|---|---|---|
| VQ-VAE (MIMI) | 8 RVQ | 12.5 | 1.1KBPS | 3.11 | 9.4% | 2.13 | 0.87 | 57.7 ± 1.3 |
| VAE | 32 DIMS | 12.5 | - | **3.15** | **8.1%** | **2.42** | **0.90** | **66.0 ± 1.4** |

**Model and dataset:** Starting from Helium-1 (Kyutai, 2025), a pretrained 2B parameters multilingual text LM as backbone, we train on French and English speech data following Défossez et al. (2024b) to learn continuation. To enhance the stability and coherence of speech continuation, we adopt the concept of *inner monologue* (Défossez et al., 2024b)—a latent textual representation of the model's own speech, aligned such that each word is positioned at the timestep corresponding to its spoken occurrence. This implies that, at each timestep $s$, the backbone transformer takes both text tokens and speech latents as input, and that its output $\mathbf{z}_{\text{long}}^s$ is passed through a linear layer which produces text logits alongside conditioning the consistency head. This internal text stream acts as a semantic scaffold, as it represents the next word to be pronounced, guiding the generation of audio tokens by grounding them in a linguistic form. Crucially, like in Défossez et al. (2024b), we introduce a temporal delay of 2 time steps (160ms) between the inner monologue and the corresponding audio tokens. This delay allows the model to access textual content prior to generating acoustic latents, decoupling high-level planning from low-level synthesis. For speech generation, we didn't notice any gains from introducing a short context transformer and noising the latents before feeding them to the backbone, resulting in a simpler model architecture.

Table 2: **Comparison of speech continuation models**: 8-RVQ RQ-transformer vs 1-step Consistency model head, with 2 temperature options.

| Model Type | Sampling temperature | Overall Speedup (↑) | Sampler Speedup (↑) | % Time in Sampler (↓) | PPX (↓) | VERT (↓) | Acoustic Quality(↑) | Meaningfulness Elo (↑) | Rank (↓) |
|---|---|---|---|---|---|---|---|---|---|
| Reference | – | – | – | – | 20.2 | 25.2 | 4.02 ± 0.11 | 2180 ± 30 | – |
| RQ-transformer 8 RVQ | 1.0 | ×1.0 | ×1.0 | 26.7% | 52.4 | 36.3 | 2.42 ± 0.12 | 1841 ± 25 | 4 |
| RQ-transformer 8 RVQ | 0.8 | ×1.0 | ×1.0 | 26.7% | 26.8 | 33.1 | 2.75 ± 0.14 | 1870 ± 30 | 3 |
| CALM - Consistency - 1 step | 1.0 | ×1.3 | ×12.3 | 2.9% | 42.9 | 34.3 | 2.82 ± 0.13 | 1947 ± 28 | 2 |
| CALM - Consistency - 1 step | 0.8 | ×1.3 | ×12.3 | 2.9% | **23.8** | **31.2** | **3.45 ± 0.14** | **2023 ± 27** | **1** |

**Results:** Tab. 1 shows that our 32-dimensional VAE matches an 8-RVQ Mimi codec on MOSNet (Lo et al., 2019), which measures audio quality, and exceeds it on the ABX metric (Schatz et al., 2013). ABX evaluates phonetic discriminability by testing whether a word like "bat" is represented closer to another "bat" utterance than to a similar-sounding word like "bit", based on latent distances. Tab. 2 shows that the 1-step Consistency model outperforms the RQ-Transformer with 8 RVQ on all our automatic and human based metrics as well as on speed. For automatic metrics, we compute PPX and VERT as introduced by Lakhotia et al. (2021). The PPX metric measures the semantic meaningfulness of the generated speech. To do so we generate 1000 excerpts of speech of 30 second, we use Whisper (Radford et al., 2022) to compute textual transcriptions and finally compute the negative log-likelihood of the text tokens with a Mistral 7B LLM Jiang et al. (2023) and convert it to Perplexity. Because a model that generates poorly diverse but good quality sentences would perform well on the PPX metric, the authors of (Lakhotia et al., 2021) introduce the VERT metric

(for diVERsiTy) which is a geometric mean of self- and auto-BLEU metrics. We use the official implementation from the fairseq Ott et al. (2019) repository.

To assess perceptual quality, we conduct two human evaluation studies involving 50 participants and 50 randomly selected examples from the English test set. Each participant rates 10 examples across the following evaluation protocols: For Acoustic Quality, participants are presented with all model continuations for the same prompt, including the ground truth reference, and rate the acoustic quality of each continuation on a 1 to 5 scale. For Meaningfulness, participants are shown two continuations of the same prompt and select the one that is the most meaningful. These pairwise preferences are used to compute an Elo score (see Sec. H).

Notably, we note a clear quality and meaningfulness improvement with our temperature method. Given that there is a text stream to guide the audio generation, we expected the CALM to match the baseline on meaningfulness rather than outperforming it. This phenomenon could be due to less model capacity in the backbone being allocated to audio manipulation, allowing more for text prediction. On the inference speed side, the consistency head is ×12.3 faster than the RQ-Transformer. The overall gain to perform a full inference of 30 seconds is ×1.3.

**Temperature Sampling and speaker similarity:** In Sec. C, we show that our gaussian temperature sampling heuristic has similar effects on speaker similarity than the temperature sampling of the discrete model.

## 5.2 TEXT-TO-SPEECH (TTS)

**VAE, model and dataset:** We use the same VAE as for speech continuation. Our TTS CALM builds on a 300M backbone transformer and uses the same architecture for the 10M parameters consistency sampling head. The text is fed to the backbone as a prefix with SentencePiece model (Kudo & Richardson, 2018) with a vocabulary size of 4k. The training data is a mix of public datasets totalizing to 88k hours of speech that is detailed in Sec. D.

**Results:** We evaluate on the Librispeech test-clean set using the same protocol as F5-TTS (Chen et al., 2025). We compare against four baselines: F5-TTS (Chen et al., 2025), DSM (Zeghidour et al., 2025), DiTAR (Jia et al., 2025), and SALAD (Turetzky et al., 2024). For DiTAR and SALAD, we report the paper results since the models are closed-source. We report Word Error Rate (WER) and Character Error Rate (CER) using Whisper-large-v3 (Radford et al., 2022), Speaker Similarity using WavLM-large (Chen et al., 2021b) as well as the results of a MUSHRA test for acoustic quality and a pairwise audio test for speaker similarity. By doing distillation on the backbone, we obtain Pocket TTS, a 100M parameters model that can run faster than real-time on a CPU (see Sec.G).

Table 3: **Text-to-Speech models.** Our CALM model with 1-step LSD outperforms baselines on WER, CER and Acoustic Quality. Results for Pocket TTS are in Sec. G.

| MODEL | NUM. PARAMETERS | WER (↓) | CER (↓) | SIM (↑) | ACOUSTIC QUALITY (↑) | SPEAKER SIM HUMAN ELO (↑) |
|---|---|---|---|---|---|---|
| REFERENCE | – | 2.23 | – | 0.69 | 61.8 ± 2.4 | 1953 ± 24 |
| REFERENCE (WITH VAE) | – | – | – | 0.57 | – | – |
| F5 TTS (NFE=32) (CHEN ET AL., 2025) | 336M | 2.42 | – | 0.66 | 54.7 ± 2.8 | 2032 ± 18 |
| DSM (16 RVQ CFG=3 | 750M | 1.95 | – | **0.67** | 60.2 ± 2.4 | **2112 ± 20** |
| W.R.T TEXT AND AUDIO PROMPT) (ZEGHIDOUR ET AL., 2025) | | | | | | |
| DiTAR (NFE=10) (JIA ET AL., 2025) | 600M | 2.39 | – | **0.67** | – | – |
| SALAD (NFE=20) (TURETZKY ET AL., 2024) | 350M | – | 0.74 | 0.54 | – | – |
| CALM W/ LSD (NFE=1, CFG=1.5 W.R.T TEXT) | **313M** | **1.81** | **0.57** | 0.52 | **61.1 ± 2.3** | 1966 ± 23 |

Our CALM model with 1-step LSD (Boffi et al., 2025) outperforms baselines on WER, CER and Acoustic Quality but obtains a low speaker similarity score. This can be partially explained by the fact that when computing the speaker similarity between the reference prompt and the reference utterance that goes through the VAE (the second line of the tab) we obtain a similarity of 0.57. Yet, we demonstrate in Tab. 1 that our VAE faithfully reconstructs speech. Due to this surprising behavior, we decide to measure speaker similarity with a human study. We observe that all measured methods beat the ground truth, which means that they preserve well the voice of the audio prompt.

### 5.3 MUSIC CONTINUATION

In this section, we detail our Music Continuation model. Since our dataset does not have any textual information, we use CLAP (Elizalde et al., 2023) to train a **text-to-music model** (see Sec. F).

**Dataset:** We use a randomly selected subset of 400K songs (approximately 20K hours with 32kHz mono format) from the LAION-Disco-12M dataset, ensuring broad coverage across musical genres.

**VAE:** Our variational autoencoder (VAE) and codec architecture is adapted from the Mimi codec (Défossez et al., 2024b), originally designed for 24kHz speech at 12.5Hz. We trained it to compress 32kHz mono music with a 25Hz frame rate. Details and metrics are described in Sec. B.

Table 4: **Comparison of music continuation for 30 seconds generation.** Consistency-based models provide up to a $2.2\times$ overall speedup and a $19.3\times$ sampler head speedup compared to the RQ-Transformer 32 RVQ baseline, with improved FAD scores and equivalent human ratings. TrigFlow achieves the best qualitative results but has significantly higher inference time. Since MusicGen only uses a linear layer to sample its token we consider its inference cost as negligible.

| MODEL | OVERALL SPEEDUP ($\uparrow$) | SAMPLER SPEEDUP ($\uparrow$) | % TIME IN SAMPLER ($\downarrow$) | FAD ($\downarrow$) | ACOUSTIC QUALITY ($\uparrow$) | ENJOYMENT ELO ($\uparrow$) | RANK ($\downarrow$) |
|---|---|---|---|---|---|---|---|
| REFERENCE | – | – | – | – | $3.84 \pm 0.08$ | $2166 \pm 33$ | - |
| RQ-TRANSFORMER 32 RVQ (BASELINE) | $\times 1.0$ | $\times 1.0$ | 57.7% | $1.06 \pm 0.06$ | $2.85 \pm 0.07$ | $1824 \pm 29$ | 4 |
| RQ-TRANSFORMER 16 RVQ | $\times 1.5$ | $\times 2.2$ | 38.0% | $1.43 \pm 0.07$ | $2.76 \pm 0.07$ | $1781 \pm 29$ | 5 |
| CALM - CONSISTENCY - 1 STEP | $\times \mathbf{2.2}$ | $\times \mathbf{19.3}$ | **6.6%** | $0.83 \pm 0.04$ | $2.90 \pm 0.07$ | $1857 \pm 28$ | 2 |
| CALM - CONSISTENCY - 4 STEPS | $\times 1.9$ | $\times 5.4$ | 20.1% | $\mathbf{0.71 \pm 0.05}$ | $3.07 \pm 0.07$ | $1847 \pm 24$ | 3 |
| CALM - TRIGFLOW - 100 STEPS | $\times 0.3$ | $\times 0.2$ | 86.6% | $\mathbf{0.64 \pm 0.04}$ | $3.12 \pm 0.07$ | $\mathbf{1921 \pm 29}$ | 1 |
| MUSICGEN MEDIUM | $\times 1.3$ | – | 0.0% | $1.72 \pm 0.12$ | $2.62 \pm 0.07$ | $1761 \pm 33$ | 6 |

**Model:** Our music CALM builds on the MusicGen Medium backbone, a 1.35B parameter Transformer (see Sec. I for all the hyperparameters). We compare against: (1) 32- and 16-RVQ discrete models utilizing the same 1.35B backbone and an RQ-transformer for parallel prediction; (2) a MusicGen Medium variant using EnCodec and delay-pattern interleaving. All baselines were trained on our dataset.

**Results:** In Tab. 4, we report both objective metrics and the results of a human evaluation study for the task of music continuation, conditioned on a 3-second prompt. We compute the speed-up compared to the RQ-Transformer 32 RVQ, the VGG Fréchet Audio Distance (FAD) on 4,000 model-generated continuations from the test set. The Acoustic Quality is a MOS score between 1 and 5. The Enjoyment metric is an Elo score (see Sec. H), computed by making human raters choose their favorite music out of generated pairs with the same 3s prompt. We observe that CALM with consistency outperforms the 32 RVQ RQ-Transformer baseline on computed and human metrics while being $\times 1.9$ to $\times 2.2$ times faster for the overall speedup. While the RQ-Transformer takes 57.7% of the inference time for the baseline, the consistency head only takes 6.6% to 20.1%. As well, we train a CALM model with a TrigFlow head instead of consistency and it outperforms all the models but to the price of a slow inference.

**Ablation Study:** In Sec. E.1, we show the importance of each architectural component.

**Necessity of Consistency for fast inference:** Sec. E.4 shows that consistency models largely outperforms TrigFlow under 10 inference steps (the regime where RTF < 1).

**Scalability:** We show in Sec. E.5 that CALM does improve with a bigger 3B parameters backbone. However, we leave a complete scalability study for future work.

## 6 CONCLUSION

We present *Continuous Audio Language Models* (CALM), a novel framework for autoregressive audio generation that operates directly in the continuous latent space of a VAE, bypassing the limitations of discrete quantization. Replacing RVQ or diffusion heads with consistency models significantly reduces inference cost while improving sample quality as shown by our experiments. Our architecture combines noise-injected long-term context and clean short-term context, implemented via a dual-transformer design. We introduce practical innovations to further improve sampling quality and training efficiency. We demonstrate the effectiveness of our approach across both speech and

music generation tasks. Our results suggest that continuous modeling offers a compelling alternative to discrete tokenization for high-quality, efficient, and scalable autoregressive audio generation.

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

## A   LAGRANGIAN SELF-DISTILLATION

Lagrangian Self-Distillation (LSD) (Boffi et al., 2025) extends the consistency-model framework by introducing an additional time parameter $s$, enabling the model to learn mappings between arbitrary points along the probability flow trajectory rather than only from noisy inputs to clean data. An LSD model is a neural network $f_\phi(\mathbf{x}, t, s)$ that predicts the state of the PF-ODE solution at time $s$ given its state $\mathbf{x}_t$ at time $t$: $\forall (s, t) \in [0, 1]^2, f_\phi(\mathbf{x}_t, t, s) = \mathbf{x_s}$. In particular, with $t = 1, s = 0$ and by sampling $\mathbf{x}_1 \sim \mathcal{N}(0, I)$ we can sample from the data distribution with $f_\phi(\mathbf{x}_1, t = 1, s = 0)$.

In (Boffi et al., 2025), the authors define $\mathbf{F}_\phi(\mathbf{x}, t, s)$ as

$$f_\phi(\mathbf{x}, t, s) = \mathbf{x} + (s - t)F_\phi(\mathbf{x}, t, s). \tag{4}$$

To train a LSD model from scratch we need to combine a flow matching loss as well as a LSD loss: $\mathcal{L} = \mathcal{L}_{FM} + \mathcal{L}_{LSD}$. With an adaptive weighting loss $w_\psi(t, s)$ and our notations they derive as:

$$\mathcal{L}_{\text{FM}}(\phi, \psi) = \mathbb{E}_{\mathbf{x}_0 \sim p_{\text{data}}, \, \boldsymbol{\epsilon} \sim \mathcal{N}(0, \mathbf{I}), \, t \sim \mathcal{U}(0,1)} \left[ e^{-w_\psi(t,t)} \left\| F_\phi(\mathbf{x}_t, t, t) - (\alpha'_t \mathbf{x}_0 + \sigma'_t \boldsymbol{\epsilon}) \right\|_2^2 + w_\psi(t, t) \right]. \tag{5}$$

and

$$\mathcal{L}_{\text{LSD}}(\phi, \psi) = \mathbb{E}_{\mathbf{x}_0, \boldsymbol{\epsilon}, t, s} \left[ e^{-w_\psi(t,s)} \left\| \partial_s f_\phi(\mathbf{x}_t, t, s) - F_{\phi^-} (f_\phi(\mathbf{x}_t, t, s), s, s) \right\|_2^2 + w_\psi(t, s) \right]. \tag{6}$$

where $\mathbf{x}_0 \sim p_{\text{data}}, \, \boldsymbol{\epsilon} \sim \mathcal{N}(0, \mathbf{I}), t \sim \mathcal{U}(0, 1), s \sim \mathcal{U}(0, t)$.

In practice, we compute the flow matching loss on 75% of a batch and the LSD loss on the 25% left.

## B   OUR MUSIC VAE

Table 5: **Music compression models.** At least 96 VAE latent dimensions are required to outperform the 32-RVQ codec on reconstruction metrics. EnCodec has been retrained on our dataset.

| MODEL TYPE | DIMS / RVQ | FRAME RATE (HZ) | BITRATE (KBPS) | VISQOL (↑) | SISNR (↑) |
|---|---|---|---|---|---|
| ENCODEC COPET ET AL. (2023) | 4 RVQ | 50 | 2.2 | 2.41 | 5.62 |
| VQ-VAE (INSPIRED FROM MIMI) | 32 RVQ | 25 | 8.8 | 3.63 | 9.61 |
| VAE | 32 DIMS | 25 | – | 2.23 | 5.51 |
| VAE | 96 DIMS | 25 | – | 3.65 | 9.76 |
| VAE | 128 DIMS | 25 | – | **4.01** | **10.3** |

Our variational autoencoder (VAE) and codec architecture is adapted from the Mimi codec (Défossez et al., 2024b), originally designed for 24kHz speech at 12.5Hz. We trained it to compress 32kHz mono music with a 25Hz frame rate. We experiment with bottleneck sizes of 96 and 128 dimensions. For comparison, MusicGen's EnCodec model (Copet et al., 2023) also operates at 32kHz but uses a 4-level RVQ at 50Hz. In Tab. 5, we report reconstruction metrics (audio ViSQOL Chinen et al. (2020) and SISNR), showing that a 32-dim VAE matches MusicGen's codec, and that at least 96 dimensions are needed for our VAE to match the quality of the 32-level RVQ configuration.

## C   EFFECT OF GAUSSIAN TEMPERATURE SAMPLING

We evaluate how our proposed gaussian temperature sampling method affects acoustic diversity for consistency models, in comparison with temperature sampling on its discrete counterpart. More precisely, we compute WavLM speaker embeddings (Chen et al., 2021b) over 100 unprompted speech generations with different values of temperature, both for an RQ-transformer and CALM. In both cases, the inner monologue text stream is generated with 0.8 temperature to ensure text diversity. We then average the pairwise cosine similarities of these embeddings, as a measure of diversity: the higher the average similarity, the lower speaker diversity in generated audio. The reference speaker similarity number is computed over 100 examples from the ground truth dataset. Fig. 2 shows, as expected, that speaker similarity tends to decrease with temperature, which means that temperature increases diversity with a similar trajectory to the discrete models.

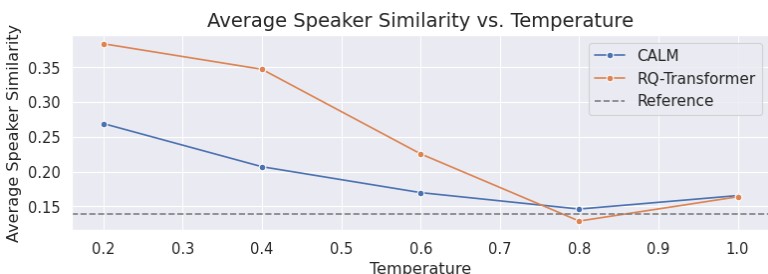

Figure 2: Average pairwise speaker similarity over 100 unprompted 10s generations, or 100 10s examples from the ground truth dataset as reference. As expected, for both methods models generate more diverse speakers (i.e. less pairwise speaker similarity) as temperature increases.

## D DATA USED FOR THE TEXT-TO-SPEECH MODEL

The dataset used to train our TTS models is composed of AMI Carletta (2007), EARNINGS22 Del Rio et al. (2022), GIGASpeech Chen et al. (2021a), SPGISpeech O'Neill et al. (2021), TED-LIUM Hernandez et al. (2018), VoxPopuli Wang et al. (2021), LibriHeavy Beck et al. (2024), and Emilia Author & Contributors (2023). It results into 88k hours of audio.

## E SUPPLEMENTARY EXPERIMENTS

### E.1 ABLATION STUDY

| MODEL VARIANT | FAD ($\downarrow$) |
|---|---|
| BASE (CALM - CONSISTENCY - 4 STEPS) | $\mathbf{0.93 \pm 0.06}$ |
| W/O HEAD BATCH MULTIPLIER | $1.32 \pm 0.09$ |
| W/O NOISE AUGMENTATION | $1.63 \pm 0.11$ |
| W/O SHORT CONTEXT TRANSFORMER | $4.03 \pm 0.16$ |
| W/O ANY OF THE ABOVE | $8.38 \pm 0.17$ |

Table 6: **Ablation study on music CALM Consistency 4-steps model components, after 250K training steps.** Removing noise augmentation or the short-context transformer leads to significant performance drops. Final row approximates the MAR configuration from Li et al. (2024).

An ablation study (Tab. 6) on music CALM Consistency 4 steps shows the importance of each component. The experiments are run for 250K steps, which explains that the base model's FAD is worse than the one reported in Tab. 4. The final row that is the closest to the MAR framework (consistency replacing diffusion) fails to produce high-quality music. In Fig. 3, we show that the FAD decreases much faster over time when we train consistency CALM models with a bigger head batch multiplier. All evaluations are done with 4 steps of consistency. We keep the value of 8 for all of our experiments as a higher value would lead to out of memory issues at training time.

### E.2 HEAD BATCH MULTIPLIER VALUE

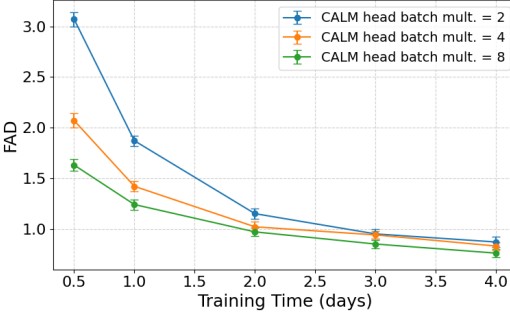

Figure 3: **Effect of the head batch multiplier value.** Training a model (music consistency CALM) with a higher batch size multiplier fastens the convergence for the FAD metric. All the evaluations are done with 4 steps of consistency at inference time.

### E.3 SHORT-CONTEXT TRANSFORMER WINDOW

To study the influence of the context window $K$ of the short-context transformer, we performed an hyperparameter search with $K = 5, 10, 20, 40$ and trained our models for 500K training steps (see Tab. 7). The model is the music continuation CALM and all evaluations are done with 4-steps of consistency. We observe that finding an optimal value of $K$ is not critical even though it seems that 5 is too small enough.

Table 7: **FAD after 500K training steps for different short-context Transformer contexts $K$.**

| $K$ in short-context Transformer | FAD |
|---|---|
| 5 | $0.85 \pm 0.05$ |
| 10 | $0.76 \pm 0.04$ |
| 20 | $0.73 \pm 0.04$ |
| 40 | $0.78 \pm 0.04$ |

### E.4 COMPARISON OF TRIGFLOW AND CONSISTENCY ON INFERENCE SPEED AND QUALITY CRITERIA

In Tab. 8 we compare the inference speed for both TrigFlow CALM and Consistency CALM on the task of music continuation on our test set. We see that below the 10 steps regime, only the Consistency model performs well. Moreover, around 10 steps is the limit for streaming applications where the real time factor (RTF) is smaller than 1. This justifies the need of consistency modeling (instead of diffusion or flow matching) for good quality streaming generation.

Table 8: **Generation efficiency of TrigFlow CALM and Consistency CALM models.** We compute the inference time to generate 30 seconds of audio, the corresponding Real Time Factor (RTF) as well as the FAD metric for different numbers of inference steps. For streaming (RTF<1), only consistency generates good quality audio.

| # Steps | Time (s) | RTF | FAD (TrigFlow) | FAD (Consistency) |
|---|---|---|---|---|
| 1 | 16.7 | 0.56 | – | $0.83 \pm 0.04$ |
| 4 | 20.4 | 0.68 | $28.83 \pm 0.20$ | $0.71 \pm 0.05$ |
| 10 | 27.7 | 0.92 | $4.62 \pm 0.07$ | $0.73 \pm 0.05$ |
| 25 | 46.1 | 1.54 | $0.79 \pm 0.04$ | $0.96 \pm 0.06$ |
| 50 | 76.8 | 2.56 | $0.74 \pm 0.05$ | $1.46 \pm 0.06$ |
| 100 | 136.4 | 4.55 | $0.64 \pm 0.04$ | $2.05 \pm 0.07$ |

### E.5 SCALABILITY OF CALM

In order to see if all the hyperparameters of our CALM method transfer well to more model parameters, we trained a consistency music CALM model with a larger 3B backbone (with a model dimension of 2048, 32 heads and 48 layers) as well as a discrete RQ-Transformer with 32-RVQ model with the same backbone size. For the CALM model, we use 4 consistency steps at inference. Tab. 9 shows that the FAD are improving with a bigger backbone (3B vs 1.3B) in similar proportions for both discrete-based (RQ-Transformer) and continuous-based (CALM) models.

Table 9: **Scalability of the backbone for CALM and RQ-Transformer methods.**

| Model | FAD |
|---|---|
| CALM with 3B backbone | $0.62 \pm 0.05$ |
| 32 RVQ RQ-Transformer with 3B backbone | $0.98 \pm 0.06$ |
| CALM with 1.3B backbone | $0.71 \pm 0.05$ |
| 32 RVQ RQ-Transformer with 1.3B backbone | $1.06 \pm 0.06$ |

We can see in Tab. 9 that at this smaller scale ($\sim$300M), Lagrangian Self-Distillation models provide better audio quality than Consistency models. We also conclude that the latent CFG has a significant impact over the WER. As well, applying the latent CFG on the audio prefix improves the speaker similarity but at the cost of audio quality. All models were run with sampling from a Gaussian with 0.7 temperature (i.e., multiplying the standard Gaussian noise by $\sqrt{0.7}$).

### E.6 TEXT-TO-SPEECH ABLATION STUDY

At the 300M parameter scale, Lagrangian Self-Distillation (LSD) achieves significantly higher audio quality than standard Consistency models. Furthermore, latent CFG has a major positive impact on Word Error Rate (WER), though applying it to the audio prefix trades overall audio quality for increased speaker similarity.

Table 10: Ablation study on the Text-to-Speech CALM model ($\sim$300M parameters). We evaluate the impact of Lagrangian Self-Distillation (LSD) versus Consistency modeling, as well as the effects of latent CFG applied to text and audio prefixes.

| MODEL | NUM. PARAMETERS | WER ($\downarrow$) | CER ($\downarrow$) | SIM ($\uparrow$) | MUSHRA ($\uparrow$) | SPEAKER SIM (HUMAN EVAL ELO ($\uparrow$)) |
|---|---|---|---|---|---|---|
| CALM W/ LSD (NFE=1, CFG=1.5 W.R.T TEXT) | 313M | 1.81 | 0.57 | 0.52 | $61.1 \pm 2.3$ | $1966 \pm 23$ |
| CALM W/ LSD (NFE=1, NO CFG) | 313M | 2.39 | 0.90 | 0.52 | $55.5 \pm 2.2$ | $1991 \pm 26$ |
| CALM W/ LSD (NFE=1, CFG=1.25 W.R.T TEXT AND AUDIO PREFIX) | 313M | 1.86 | 0.59 | 0.54 | $56.2 \pm 2.2$ | $1995 \pm 27$ |
| CALM W/ CONSISTENCY (NFE=1, CFG=1.5 W.R.T TEXT) | 313M | 1.93 | 0.62 | 0.40 | $46.8 \pm 2.5$ | $1900 \pm 30$ |

## F TEXT-TO-MUSIC GENERATION

We compare CALM on the task of text-to-music generation. Since our dataset do not have any text labeling, we use CLAP (Elizalde et al., 2023) as a prefix conditioning and train MusicGen, our RQ-Transformer 32 RVQ baseline as well as CALM. During training, we drop the conditioning 20% of the time in order to perform Classifier Free Guidance (Ho & Salimans, 2022) at inference. We use CLAP in audio mode when training. We use the same training data, VAE/codecs and hyperparameters as the models reported in Tab. 4. In Tab. 11, we show the results of our CLAP conditioned model evaluated on our internal test set with the CLAP conditioning being used in audio mode. As well, we report the results of our model and several baselines on the text-to-music benchmark on the MusicCaps dataset Agostinelli et al. (2023) in Sec.F.

Table 11: **Text-to-music generation models** on our test set where CLAP is used in audio mode.

| MODEL | OVERALL SPEEDUP ($\uparrow$) | SAMPLER SPEEDUP ($\uparrow$) | % TIME IN SAMPLER ($\downarrow$) | FAD ($\downarrow$) | ACOUSTIC QUALITY ($\uparrow$) | ENJOYMENT ELO ($\uparrow$) | RANK ($\downarrow$) |
|---|---|---|---|---|---|---|---|
| REFERENCE | – | – | – | – | $3.75 \pm 0.15$ | $2104 \pm 24$ | – |
| RQ-TRANSFORMER 32 RVQ (CFG=3) | $\times 1.0$ | $\times 1.0$ | 57.7% | $0.93 \pm 0.07$ | $3.16 \pm 0.15$ | $1998 \pm 24$ | 1 |
| CALM - CONSISTENCY - 4 STEPS (CFG=2) | $\times 1.9$ | $\times 5.4$ | 20.1% | $0.91 \pm 0.08$ | $3.11 \pm 0.14$ | $1998 \pm 24$ | 1 |
| MUSICGEN MEDIUM | $\times 1.3$ | – | 0.0% | $1.93 \pm 0.14$ | $2.54 \pm 0.14$ | $1946 \pm 25$ | 3 |

We report the results on the MusicCaps dataset Agostinelli et al. (2023) and report as well the results of MusicLM (Agostinelli et al., 2023), the original text-to-music MusicGen Medium (Copet et al., 2023), AudioLDM2 (Liu et al., 2024), Noise2Music (Huang et al., 2023), Jen-1 (Li et al., 2025) as well as MusicFlow (Prajwal et al., 2024) on the FAD, the KL Divergence (KLD) metric based on the pre-trained audio model PANN (Kong et al., 2019) as well as the CLAP cosine similarity that measures the matching between the generated music and text description. The results of Tab. 12 show that even though our goal is not to specifically build the best text-to-music model, simply applying CALM on our dataset with a CLAP conditioning leads to competitive results. The metrics from the models that we did not train are reported from the MusicFlow (Prajwal et al., 2024) paper.

## G POCKET TTS

Given a 313M parameters text-to-speech CALM model that has a 24 layers backbone transformer (the teacher), we do latent distillation to a 6 layers transformer backbone (the student) and a CFG coefficient of $\alpha = 1.5$ applied to the teacher. We keep the same MLP sampling head. The final model,

Table 12: Text-to-music results on the MusicCaps dataset.

| MODEL | FAD ($\downarrow$) | KLD ($\downarrow$) | CLAP ($\uparrow$) |
|---|---|---|---|
| REFERENCE | – | – | 0.30 |
| MUSICLM (AGOSTINELLI ET AL., 2023) | 4.00 | – | – |
| MUSICGEN MEDIUM (COPET ET AL., 2023) | 3.40 | 1.23 | 0.37 |
| AUDIOLDM2 (LIU ET AL., 2024) | 3.13 | 1.20 | 0.43 |
| NOISE2MUSIC (HUANG ET AL., 2023) | 2.10 | – | – |
| JEN-1 (LI ET AL., 2025) | 2.00 | 1.29 | – |
| MUSICFLOW (UNIDIRECTIONAL LM + FM) (PRAJWAL ET AL., 2024) | 2.69 | 1.23 | 0.52 |
| MUSICGEN MEDIUM (retrained with CLAP) | 2.70 | 1.37 | 0.39 |
| RQ-TRANSFORMER 32 RVQ | 2.56 | 1.35 | 0.43 |
| CALM - CONSISTENCY - 4 STEPS | 2.14 | 1.30 | 0.44 |

named **Pocket TTS** has a size of 90M parameters while the VAE has 20M parameters. In Tab.13, we put 100M parameters to include the VAE decoder in the parameter count.

We evaluate Pocket TTS on the Librispeech test-clean set following the same protocol as F5-TTS (Chen et al., 2025), with the difference that we cleaned the audio input using Adobe Enhance Speech[1] to obtain 24kHz high-quality audio. We evaluate all baselines with the enhanced samples[2].

We compare against three baselines: F5-TTS (Chen et al., 2025), DSM (Zeghidour et al., 2025), Chatterbox Turbo[3] and Kokoro TTS[4]. We report Word Error Rate (WER) using Whisper-large-v3(Radford et al., 2022), as well as the results of a pairwise human evaluation for audio quality and speaker similarity. For audio quality, we ask raters "Which of the two audio clips has the best audio quality?", and for speaker similarity, we ask "Which of the two audio clips sounds more similar to the reference audio clip in terms of voice characteristics?" and provide the voice prompt as a reference.

Table 13: Comparison of TTS models on Librispeech test-clean.

| MODEL | PARAM SIZE (GEN. ONLY) | WER ($\downarrow$) | AUDIO QUALITY (ELO) ($\uparrow$) | SPEAKER SIM (ELO) ($\uparrow$) | FASTER THAN REAL-TIME CPU |
|---|---|---|---|---|---|
| F5-TTS (CHEN ET AL., 2025) | 336M | 2.21 | $1949 \pm 27$ | $1946 \pm 26$ | $\times$ |
| DSM | 750M | 1.84 | $1959 \pm 25$ | $2037 \pm 21$ | $\times$ |
| CHATTERBOX TURBO | 350M | 3.24 | $2055 \pm 23$ | $2012 \pm 22$ | $\times$ |
| KOKORO | 82M | 1.93 | NO VOICE CLONING | | $\checkmark$ |
| **POCKET TTS (OURS)** | **100M** | **1.84** | $2016 \pm 25$ | $1898 \pm 26$ | $\checkmark$ |

As seen in Tab. 13, Pocket TTS has the lowest Word Error Rate, a better Audio Quality than the ground truth F5-TTS and DSM, as well as an on-par Speaker Similarity with the ground truth while being a significantly smaller model than competitors, and being the only one that can run faster than real-time on CPU (we tested on Apple M3 and Intel core ultra 7 165H). We invite the reader to check the blog post where the one-line installation of the model is provided: kyutai.org/pocket-tts-technical-report.

## H   HUMAN EVALUATION METHODS

Audio clips are always 30s second total, with a 3s prompt coming from a ground truth audio. Each experiments has 50 samples for each method. There are 50 raters. Each of them sees 10 samples. Raters were paid £9 / hour for their contribution.

---

[1] https://podcast.adobe.com/en/enhance

[2] https://huggingface.co/datasets/kyutai/librispeech_test_clean_enhanced

[3] https://www.resemble.ai/chatterbox-turbo/

[4] https://huggingface.co/hexgrad/Kokoro-82M

## H.1 SPEECH CONTINUATION

**Acoustic quality assessment**: How would you rate the overall quality of this audio clip? Consider aspects such as clarity, balance, richness, and naturalness. Listen to at least 10 seconds of audio before deciding. 1 clip is presented, possibilities are bad, poor, fair, good, excellent.

**Meaningfulness**: Which of these two audio clips feels more like meaningful and natural speech? The first 3 seconds are identical. Listen to at least 10 seconds of each clip. 2 clips are presented, ties are possible. Elo scores are Bayesian estimates of the posterior mean in a Bradley-Terry model.

## H.2 MUSIC CONTINUATION AND CLAP-TO-MUSIC

**Acoustic quality assessment**: How would you rate the overall quality of this music? Consider aspects such as clarity, balance, richness, and naturalness. Listen to at least 10 seconds of audio before deciding. 1 clip is presented, possibilities are bad, poor, fair, good, excellent.

**Music enjoyment**: Which music do you enjoy listening to more? The first 3 seconds are identical. Listen to at least 10 seconds of each clip. 2 clips are presented, ties are possible. Elo scores are Bayesian estimates of the posterior mean in a Bradley-Terry model.

## H.3 BAYESIAN ELO SCORE

The Meaningfulness metric for speech and the Enjoyment metric for music are both Bayesian Elo Scores. Elo score is used to rank models based on some pairwise comparisons of audio samples. Given two models $A$ and $B$, the probability that $A$ is preferred over $B$ is:

$$P(A > B) = \frac{1}{1 + 10^{(E_B - E_A)/400}} \quad (7)$$

where $E_A$ and $E_B$ are the Elo scores of each model. Unlike a traditional Elo score, the Bayesian Elo score uses a Gamma prior, so that one can derive confidence intervals over the posterior distribution.

By defining $S_A$ such as $E_A = 400 \log_{10}(S_A) + c$ with $c$ being a constant, we obtain:

$$P(A > B) = \frac{S_A}{S_A + S_B} \quad (8)$$

which is a Bradley-Terry (Bradley & Terry, 1952) model. There are a few different methods to estimate the parameters of a Bradley-Terry model. We use the iterative one from (Caron & Doucet, 2010) where $S_A^0$ follows a Gamma prior with parameters $\alpha^0, \beta^0$. By denoting $w_A$ the number of times where method $A$ won against any other methods and $n_{AB}$ the number of times where $A$ and $B$ are compared, $S_A^t$ is computed with the following update rule until convergence:

$$S_A^{t+1} = \frac{\alpha + w_A}{\beta + \sum_{B \neq A} \frac{n_{AB}}{S_A^t + S_B^t}} \quad (9)$$

which is the mean of the Gamma distribution with updated parameters $\alpha_A^{t+1}, \beta_A^{t+1}$:

$$\alpha_A^{t+1} = \alpha^0 + w_A, \quad \text{and} \quad \beta_A^{t+1} = \beta^0 + \sum_{B \neq A} \frac{w_{A,B}}{S_A^t + S_B^t}. \quad (10)$$

Iterating over $t$ allows to reach a fix point, we run 30 of them, once we have collected all the pairs. We use $\alpha = 0.1, \beta = 0.1, c = 2000$ so that in absence of any data, $S_A = 1$ and $E_A = 2000$. Confidence interval are 95% confidence interval according to the posterior (the 2.5th and 97.5th percentiles).

## I  HYPERPARAMETERS

We present in Tab. 14 the parameters of the music and speech VAE used for CALMs. For CALMs and the RQ-Transformer based discrete LMs, the hyperparameters are presented in Tab. 15.

Table 14: VAE hyperparameters

| | Music VAE | Speech continuation VAE |
|---|---|---|
| *General* | | |
| Sample rate | 32kHz | 24kHz |
| Frame rate | 25Hz | 12.5Hz |
| Latent dimension | 128 | 32 |
| *Architecture* | | |
| Convolutions ratios | 8, 8, 5, 4 | 6, 5, 4, 4, 4 |
| Num transformer encoder layers | 4 | 8 |
| Num transformer decoder layers | 4 | 8 |
| Transformer context | 30s | 10s |
| *Training parameters* | | |
| Batch size | 64 | 64 |
| Audio sample length | 12s | 12s |
| KL loss weight | 0.01 | 0.01 |
| Reconstruction loss | ✓ | ✗ |
| Distillation loss weight | ✗ | 25 |
| LR Schedule | cosine | cosine |
| Learning rate | $8 \cdot 10^{-4}$ | $8 \cdot 10^{-4}$ |

Table 15: Model and training hyperparameters

| | Music | Speech continuation | Text-to-Speech |
|---|---|---|---|
| *Backbone Transformer* | | | |
| Model dimension | 1536 | 2560 | 1024 |
| MLP dimension | 6336 | 10560 | 4096 |
| Number of heads | 24 | 20 | 16 |
| Number of layers | 48 | 24 | 24 |
| Learning rate | $1 \cdot 10^{-4}$ | $5 \cdot 10^{-5}$ | 1e-4 |
| Number of parameters | 1.35B | 2.2B | 302M |
| *RQ-Transformer head (RVQ model)* | | | |
| Model dimension | 1024 | 1024 | ✗ |
| MLP dimension | 4096 | 4096 | ✗ |
| Number of heads | 16 | 16 | ✗ |
| Number of layers | 6 | 6 | ✗ |
| Number of parameters | 701M | 701M | ✗ |
| *Consistency sampler head (ours)* | | | |
| Number of layers | 12 | 6 | 6 |
| MLP dimension | 3072 | 512 | 512 |
| Gating | SiLU | SiLU | SiLU |
| Number of parameters | 601M | 10M | 10M |
| *Short context transformer (ours)* | | | |
| Model dimension | 1536 | ✗ | ✗ |
| MLP dimension | 6336 | ✗ | ✗ |
| Number of heads | 24 | ✗ | ✗ |
| Number of layers | 4 | ✗ | ✗ |
| Context | 10 | ✗ | ✗ |
| Number of parameters | 113M | ✗ | ✗ |
| *Audio embedding manipulations* | | | |
| Center and normalize | ✓ | ✓ | ✓ |
| Noise before entering backbone | ✓ | ✗ | ✗ |
| *Training parameters* | | | |
| Head batch multiplier | 8 | 8 | 8 |
| Optimizer | AdamW $\beta_1 = 0.9, \beta_2 = 0.95$ | AdamW $\beta_1 = 0.9, \beta_2 = 0.95$ | AdamW $\beta_1 = 0.9, \beta_2 = 0.95$ |
| Batch size | 48 | 144 | 128 |
| Audio sample length | 30s | 300s | 60s |
| LR Schedule | cosine | cosine | cosine |
| Learning rate | $1 \cdot 10^{-4}$ | $2 \cdot 10^{-4}$ | 1e-4 |
| GPU used | 16 H100 | 48 H100 | 8 H100 |
| Number of training steps | 500k | 150k | 400k |
| Initial checkpoint | ✗ | Helium-1 (Kyutai, 2025) | ✗ |
| Inner monologue | ✗ | ✓ | ✗ |
| Acoustic delay | - | 2 frames (160 ms) | - |

