# OpenReview forum: "Continuous Audio Language Models"
_ICLR.cc/2026/Conference — ICLR 2026 Poster_

### Official Review · Reviewer_ZVTy · 2025-10-23

**Soundness:** 1
**Presentation:** 3
**Contribution:** 2
**Rating:** 4
**Confidence:** 4

**Summary:**

Discrete-token audio LMs (e.g., RVQ stacks + AR decoders) trade fidelity for latency: deeper codebooks mean more tokens and slower sampling. The paper investigates whether we can get rid of discretization and instead do autoregressive modeling in a continuous latent space while keeping quality and reducing inference cost by few-step sampling with consistency models.
The paper contributes CALM, an autoregressive framework that operates directly on continuous VAE latents and replaces RVQ/diffusion heads with a small consistency-model MLP to enable few-step next-latent prediction. This avoids discretization while targeting real-time sampling. It introduces a dual-context design, a noise-injected long-term backbone combined with a clean short-context transformer over the last K latent, whose features are summed to serve as the condition of the head. Empirically, the consistency head reduces inference cost while maintaining or improving quality.

**Strengths:**

At a high level, the paper’s short-context pathway is analogous to LocDiT in DiTAR: both introduce an additional local/short-term module on top of a long-range autoregressive backbone to recover fine detail that a purely long-context model tends to miss. That said, the mechanisms and constraints differ: the proposed short-context transformer is causal and consumes a clean recent window whose features are summed with a noised long-context stream to condition a few-step consistency head. By contrast, LocDiT performs bidirectional, intra-patch diffusion with a small prefix of historical patches, and typically runs for many denoising steps. The goals align, but the causality, noise-injection, and sampling budgets (few-step consistency vs. multi-step diffusion) are importantly different.

**Weaknesses:**

- The paper targets continuation on speech and music, but the evaluation scope is too narrow to position the method within the broader speech modality. In practice, TTS is the most common task for audio generation models. Without a TTS evaluation, it’s difficult to compare against widely used or state-of-the-art systems. Including comparisons with strong baselines (e.g., GLM-4-Voice, DiTAR) on overlapping tasks would be required. In short, the method is well-substantiated for continuation, but its position relative to on-the-shelf speech models is unclear. Adding TTS (with direct SOTA comparisons) would make the contribution much easier to compare and significantly strengthen the paper.

- Lack of references: Suggest citing IMPACT [A] in line 53, which is also a MAR-style audio generation model.

[A] Huang, K. P., Yang, S. W., PHAN, H., Lu, B. R., Kim, B., Macha, S., ... & Wang, C. IMPACT: Iterative Mask-based Parallel Decoding for Text-to-Audio Generation with Diffusion Modeling. ICML 2025

**Questions:**

Why sample $x_t$ with $x_t = cos(t)x + sin(t) \epsilon$ instead of $x_t = (1-t) \cdot x+t \cdot \epsilon$ ?

---

> ### Author Response · Authors · 2025-11-21
> **Answer to Reviewer ZVTy 1/2**
>
> We thank the reviewer for their comments and suggestions. We resubmitted an extended version of the paper with **text-to-speech and text-to-music** models that we have run since the submission with comparison to external baselines. We invite the reviewer to read the updates that are **highlighted in blue**.
> We address the weaknesses in the following points:
>
> ---
>
> ## Answer to the first weakness  ➜ (Sec.4.6, Sec.5.2, Sec.5.4, Sec.E.6)
>
> Following the reviewer's comment on the fact that we only focus on continuation, we ran experiments in the setups of **text-to-music and text-to-speech**. However, Classifier Free Guidance is detrimental to good quality generation for conditioned generative models. Since CFG cannot be applied on the trajectory of 1-step consistency models, we decide to apply the CFG on the outputs of the Backbone and Short-Context transformers.
> This method is named **Latent Classifier Free Guidance** and is described in the updated version of the paper in **Sec.4.6**.
> Note that this method has been used as well in the video-to-audio model Soundreactor [1].
>
> ---
>
> ### Text-to-Speech (TTS)  — **See Sec.5.2, Tab.3**
>
> We built a text-to-speech model with the CALM framework (**See Sec.5.2**) with consistency as well as **Lagrangian Self-Distillation (LSD)**, which is a more recent and more stable framework of 1-step flow-matching models introduced in [1].
>
> We compare our model to baselines (see **Tab.3**) **F5TTS [2], DSM [3], DiTAR [4], SALAD [5]**. Our CALM–LSD model **beats the baselines** (including DiTAR) on WER, CER, acoustic quality metrics (human study) and model size.
>
> However, it has a lower speaker similarity, which we believe is caused by our VAE. Despite having good reconstruction quality (**Tab.1**), our VAE has a low measured speaker similarity with the reference speech.
> A human study shows that our model is still preferred compared to reference speech in terms of similarity.
>
> **We invite the reviewer to listen to the samples that will be added on our website iclr-continuous-audio-language-models.github.io in the next 24h.**
>
>
> ### DiTAR vs CALM–LSD (speed / architecture differences)
> We would like to emphasize that our model is faster to sample than DiTAR on the head size:
> - DiTAR operates at a higher framerate (40Hz vs 12.5Hz) but uses a patch size of 4. The backbone framerate becomes 10Hz.
> - Their diffusion head is a **75M parameter Diffusion Transformer**, NFE=10, computing 12 tokens in parallel (2 historical patches × 4 tokens).
> ⟶ **This means that to generate 1 second of speech, DiTAR must call this 75M-parameter transformer 100 times on sequences of 12 tokens for the head**.
>
> - **CALM**, by contrast, uses a **10M-parameter MLP head**, with **NFE=1**, with a backbone operating at **12.5 Hz**.
> ⟶**This means that to generate 1 second of speech, CALM only does 12.5 passes through a 10M parameter MLP for the head**
>
> ---
>
> ### Text-to-Music (T2M) — **Sec.5.4, Sec.E.6, Tab.11**
> We trained a text-to-music model to show that CALM can perform text-conditioned music generation.
> Since our dataset does not have textual descriptions, we use a pretrained **CLAP** model as a conditioner. CLAP embeddings are computed from the audio at training time.
>
> We trained **3 models with CLAP conditioning**: MusicGen medium, RQ-Transformer baseline (32 RVQs) and CALM in consistency mode.
>
> We perform **2 evaluations**:
>
> 1. **Internal test set** using CLAP in audio mode (**see Sec.5.4**)  on which CALM and the 32RVQ baseline RQ-Transformer model are on par both on the FAD as well as on human metrics (acoustic quality and enjoyment).
> 2. **MusicCaps benchmark** using textual descriptions at inference time (**see Sec.E.6**). On this benchmark, we notice that even if our goal is not to build the best text-to-music model, **it is still competitive** with other models such as MusicFlow, Jen-1, Noise2Music, AudioLDM2, MusicGen and MusicLM (Tab.11).  Indeed, we believe that having at least a few text annotations in our music dataset would benefit text-to-music performance because of the alignment gap in CLAP.
>
> ---
>
> ## Answer to the second weakness
>
> We have added your suggestion and now cite **IMPACT** at **L146** of our paper.
>
> ---
>
> [1] Boffi et al., *How to build a consistency model: Learning flow maps via self-distillation*
> [2] Chen et al., *F5-TTS: A Fairytaler that Fakes Fluent and Faithful Speech with Flow Matching*
> [3] Zeghidour et al., *Streaming Sequence-to-Sequence Learning with Delayed Streams Modeling*
> [4] Jia et al., *DiTAR: Diffusion Transformer Autoregressive Modeling for Speech Generation*
> [5] Turetzki et al., *Continuous Speech Synthesis using per-token Latent Diffusion*

---

> ### Author Response · Authors · 2025-11-21
> **Answer to Reviewer ZVTy 2/2**
>
> ## Answer to the question
>
> For our consistency model we use the **Trigflow** formulation instead of the classic flow matching formulation because we follow the recommendation of the paper *“Simplifying, Stabilizing and Scaling Continuous-Time Consistency Models”* by Lu and Song.
>
> The authors argue that:
>
> > “Since TrigFlow is a special case of flow matching and simultaneously satisfies EDM principles, it combines the advantages of both formulations while allowing the diffusion process, diffusion model parameterization, the Probability Flow-ODE, the diffusion training objective, and the Consistency Model parameterization to all have simple expressions.”
>
> During our preliminary experiments, we noticed a slight improvement with TrigFlow for music generation and decided to keep it.  However, the difference is of minor significance.
> Finally, we think that **Lagrangian Self-Distillation [1]** seems very promising, as our TTS experiments show.
>
>
> [1] Boffi et al., *How to build a consistency model: Learning flow maps via self-distillation*

---

### Official Review · Reviewer_vAFX · 2025-10-30

**Soundness:** 3
**Presentation:** 3
**Contribution:** 2
**Rating:** 6
**Confidence:** 3

**Summary:**

To address the quality and efficiency issues of discrete tokens, the paper proposes CALM. CALM models directly in the continuous latent space and introduces a consistency model to enhance audio reconstruction quality while significantly improving inference speed. By using the Causal Short-Context Transformer module, the quality of generation is enhanced. In addition, the paper proposes practical techniques such as temperature sampling and Head Batch Multiplier to enhance training stability and controllability of generation.

**Strengths:**

The paper makes several improvements to existing Audio Language Models, introducing new mechanisms and training methods with a good degree of innovation. The paper is well-written, with the methods section presenting a complete mathematical process and the experimental section validating multiple improvements. CALM is of considerable importance to current audio language models.

**Weaknesses:**

1. Although a detailed comparison is made with the RQ-Transformer, there is insufficient comparison with other continuous audio generation models (such as SALAD, DiTAR, etc.).

2. As shown in the experimental data presented by the authors (Table 5), the Short-Context Transformer is crucial for generation quality, but the authors do not specifically explain the working mechanism of this module, nor do they explain how the absence of this module would degrade the model's generation results.

3. Similarly, the authors need to more clearly explain the specific impact mechanism of noise injection on model robustness and preferably provide some visual results.

**Questions:**

1. Could the authors provide a more detailed explanation of the working mechanism of the Short-Context Transformer, or validate through experiments and visual results what specific information this module captures?

2. Recently, some new continuous modeling-based audio language models have emerged, such as DiTAR and Ming-UniAudio. If possible, could the authors compare the technical similarities and differences as well as the actual performance between CALM and their models?


DiTAR: https://arxiv.org/abs/2502.03930

Ming-UniAudio: https://mdn.alipayobjects.com/cto_asrtts/uri/file/as/TR-Ming-UniAudio.pdf

---

> ### Author Response · Authors · 2025-11-21
> **Response to Reviewer vAFX 1/2**
>
> We thank the reviewer for their comments and suggestions. We resubmitted an extended version of the paper with **text-to-speech and text-to-music** models that we have run since the submission with comparison to external baselines. We invite the reviewer to read the updates that are **highlighted in blue**.
> We address the weaknesses in the following points:
>
> ---
>
> ## **Response to the first weakness**
>
> For the paper submission, we focused on the task of speech continuation. However, following your remark, we trained a **text-to-speech CALM** model that we added in **Sec. 5.2** of the paper, in order to compare ourselves with **SALAD** and **DiTAR** as well as **F5-TTS** and **DSM**.
>
> To perform conditioned generation with CALM, we introduce **Latent Classifier Free Guidance** in **Sec. 4.6**.
>
> For TTS, we trained CALM with both **consistency** and **Lagrangian Self-Distillation (LSD)**, a more recent and stable 1-step flow-matching framework introduced in **[1]**.
> Our **TTS CALM–LSD** model beats the baselines on **WER**, **CER**, **acoustic quality metrics** (human study), and **model size**. However, it suffers from lower speaker similarity, which we believe is due to our VAE. Indeed, even though our VAE has strong reconstruction quality (see **Tab. 1**), it still shows low measured similarity to the reference speech.
>
> However, a human study shows that our model is still preferred compared to the reference in terms of speaker similarity. We invite the reviewer to listen to the samples that will be added to our website **iclr-continuous-audio-language-models.github.io** in the next 24 hours.
>
> ---
>
> ## **Response to the second and third weakness**
>
> As noticed by the reviewer, **noise injection** and the **Short-Context Transformer (SCT)** are complementary and necessary for generating music that remains consistent over time (i.e., the quality does not degrade during long generations).
>
> Regarding the **Short-Context Transformer (SCT)**, its mechanism is similar to any autoregressive transformer, but it is **much smaller** than the backbone and operates with a **context window of only 10 frames** (0.4 seconds of audio).
> During training, the backbone sees **noisy** data, while the SCT sees **clean** data. This allows the backbone to extract coarse and long-term information, while the SCT provides recent and clean information to the MLP head, which predicts the next clean latent.
> Noise injection also makes the backbone robust to out-of-distribution inputs.
>
> At inference time, we observe that it is **not necessary to noise** the backbone inputs, and we believe this mechanism helps **reduce error accumulation**.
>
> Our **ablation study** confirms that **both components must be used together**. We invite the reviewer to listen to the audio samples on the webpage (ablation study section):
>
> - **No SCT and no noise injection** (3rd column):
>   The model sees out-of-training-distribution latent vectors and becomes unstable. Error accumulation occurs, and the music quickly degrades into distorted noise (see the 2nd example of the 3rd column).
>
> - **Noise injection but no SCT** (4th column):
>   Injecting noise during training increases robustness and prevents divergence. However, noising erases fine details, and the model cannot generate them, leading to degraded quality over long generations as well as only generating the rhythmic elements (see 1st example of the 4th column).
>
> - **SCT with no noise injection** (5th column):
>   The generation quality improves, but error accumulation may still appear (see the 3rd example of the 5th column). We believe than event though there is no noise injection, the backbone extracts only coarse information while the SCT focuses on details.
>
> Because of quantization, **RVQ token-based models** see a limited range of input tokens, unlike CALM, which operates over **continuous inputs**. We think that it explains our observation that token-based models tend to suffer less from error accumulation.
>
> ---
>
> ## **Response to the first question**
>
> We are not sure how to visually show the impact of the **Short-Context Transformer** and **noise injection**, but listening to the ablation samples on the project webpage makes the phenomenon clear. We can discuss this further if needed.
>
>
> ---
>
> ### **References**
>
> [1] Boffi et al. *How to Build a Consistency Model: Learning Flow Maps via Self-Distillation*

---

> ### Author Response · Authors · 2025-11-21
> **Response to Reviewer vAFX 2/2**
>
> ## **Response to the second question**
>
> In the paper, we present only the tasks of **music continuation** and **speech continuation**.
> However, since the submission we have also worked on **text-to-speech (TTS)**, allowing us to compare CALM to other TTS models.
> We compare our TTS model with **DiTAR** on the **LibriSpeech-clean** benchmark in **Tab. 3**.
>
> Regarding architectural differences:
>
> - **DiTAR** operates at a higher frame rate (40 Hz vs. 12.5 Hz) but uses a patch size of 4, reducing its backbone frame rate to 10 Hz.
>   Its diffusion head is a **75M-parameter Diffusion Transformer**, with **NFE=10**, sampling **12 tokens in parallel** (2 historical patches of 4).
>   ⟶ **To generate 1 second of speech, DiTAR must call this 75M-parameter transformer 100 times on sequences of 12 tokens**.
>
> - **CALM**, by contrast, uses a **10M-parameter MLP head**, with **NFE=1**, with a backbone operating at **12.5 Hz**.
>   ⟶ **To generate 1 second of speech, CALM only requires 12.5 forward passes through a 10M MLP, making generation much faster**.
>
> Regarding **Ming-UniAudio**, it is a **16.8B parameters** (2.8B active) continuous speech language model at 50 Hz. It uses a 1.35B parameters continuous audio tokenizer at 50Hz (which is almost the size of our speech LM). We think that this model size makes it hard to compare with ours but we have included it in our related work since it is nice to see that continuous speech language models can scale.
>
> However, Ming-UniAudio evaluates TTS on **Seed-en**, and we will report our newly trained TTS metrics on this dataset.
>
> Finally, we kindly emphasize that Ming-UniAudio was published in **October 2025**, and the ICLR guidelines specify that *“if a paper was published on or after July 24, 2025, authors are not required to compare their own work to that paper”* (see https://iclr.cc/Conferences/2026/ReviewerGuide#%23faq).

---

### Official Review · Reviewer_6NWz · 2025-10-30

**Soundness:** 3
**Presentation:** 4
**Contribution:** 4
**Rating:** 6
**Confidence:** 4

**Summary:**

The paper proposes CALM, a continuous autoregressive audio generation framework without discrete quantization. The idea is to first replace lossy RVQ codecs with a standard VAE, and then generate latent vectors frame-by-frame using transformers with consistency heads. Experiments show that on both speech and music continuation tasks, CALM can generate high-quality, long-context and meaningful audio contents within real-time.

**Strengths:**

1. Different from previous MAR-style work designed for single tasks like TTS and musical audio generation, CALM aims to build **a unified framework** for multiple audio generation tasks. The authors illustrated the effectiveness of CALM on **two challenging tasks**: speech and music continuation, demonstrating CALM’s ability to generate content with high semantic meaningfulness and acoustic quality.
2. The idea to replace sampler/diffusion/flow matching heads with consistency head is novel. This approach, along with the newly-proposed training strategy of head batch multiplier, significantly reduces inference and training cost.
3. The paper proposes an effective solution to the error accumulation problem in continuous audio generation with noise augmentation and short context transformer.

**Weaknesses:**

1. Although the meaningfulness and enjoyment improves upon baseline methods, the generated audio still lacks long-content semantic continuity as illustrated by samples in the demo page.
2. The proposed method is evaluated only on unconditional generation tasks, making it unclear whether the proposed method can be extended to more applicable conditioned tasks like TTS and text-to-music. This also makes it difficult to compare CALM with mainstream baselines that work on conditional generation tasks.

**Questions:**

1. Have the authors considered methods that directly generates mel-spectrogram, like MELLE [1] and Flow-Omni [2]? What is the advantage of CALM compared to these solutions?
2. How much data is used to train the speech generation model? Have the authors used in-house data? If not, the source of the training data should be provided.
3. As the reconstruction quality of VAE sets an upper bound for generated audio quality, it would be better if more detailed evaluation could be provided. For example, reporting additional metrics like PESQ and STOI, as well as perceptual metrics like MOS.

**References**

[1] Meng, Lingwei, et al. "Autoregressive speech synthesis without vector quantization." *arXiv preprint arXiv:2407.08551* (2024).

[2] Yuan, Ze, et al. "Continuous speech tokens makes llms robust multi-modality learners." *arXiv preprint arXiv:2412.04917* (2024).

---

> ### Author Response · Authors · 2025-11-21
> **Response to Reviewer 6NWz**
>
> We thank the reviewer for their comments and suggestions. We resubmitted an extended version of the paper with **text-to-speech and text-to-music** models that we have run since the submission with comparison to external baselines. We invite the reviewer to read the updates that are **highlighted in blue**.
> We address the weaknesses in the following points:
>
> ---
>
> ## **Response to the first weakness**
>
> We agree with the reviewer that although meaningfulness (for speech) and enjoyment (for music) are improved in comparison to our baselines, the generated audio still lacks semanticity. For speech, we would like to emphasize that this is a relatively small (for a speech LM) 2B-parameter model that has only undergone a pretraining phase. However, the fact that it produces sentences slightly more coherent than the RVQ-based model, with similar audio quality and lower inference time is encouraging for building speech LMs in the future.
>
>
> ---
>
> ## **Response to the second weakness**
>
> Following the reviewer’s comment that we only focus on unconditional generation tasks, we ran experiments in the **text-to-music** and **text-to-speech** setups.
>
> However, **Classifier-Free Guidance (CFG)** is detrimental to high-quality generation for conditioned generative models. Since CFG cannot be applied on the trajectory of **1-step consistency models**, we apply CFG on the outputs of the Backbone and Short-Context transformers instead. This method is called **Latent Classifier-Free Guidance**, and it is described in the updated paper (see **Sec. 4.6**). Note that this method is also used in the video-to-audio model **Soundreactor [1]**.
>
> ### **Text-to-music**
> Since our dataset does not include textual descriptions, we use a pretrained CLAP model as conditioner. We specifically use the open-source **LAION music CLAP** as prefix conditioning. We compute CLAP embeddings from the audio during training.
>
> We trained three CLAP-conditioned models which are MusicGen Medium,  our RQ-Transformer baseline (32 RVQs), and CALM in consistency mode.
>
> We perform two evaluations:
> 1. On our internal test set using **audio-mode CLAP** (see **Sec. 5.4**).  In this setup, the RQ-Transformer baseline and CALM are very close (same as for continuation)
> 2. On **MusicCaps**, using textual descriptions at inference (see **Sec. E.6**), we observe that even though our goal is not to build the best text-to-music model, our system remains competitive with MusicFlow, MusicLM, MusicGen, AudioLDM2, Noise2Music and Jen-1 (see **Tab. 11**). We believe that having even a small number of text annotations in our music dataset would improve performance of our generative models.
>
> ### **Text-to-speech**
> We built a text-to-speech model with the **CALM framework** (see **Sec. 5.2**), using both **consistency**, and  **Lagrangian Self-Distillation (LSD)**, a more recent and stable 1-step flow-matching method introduced in **[2]**.
>
> Our **CALM-LSD** model beats baselines on **WER**, **CER**, **acoustic quality metrics** (human study), and **model size (see Tab. 3)**. However, it suffers from lower speaker similarity, likely caused by our VAE. Although our VAE has strong reconstruction quality (see **Tab. 1**), it still shows low speaker similarity to the reference speech (second line of Tab.3). However, the speaker similarity measured by humans (last column of Tab.3) shows that our model is still preferred in terms of similarity compared to the reference. We invite the reviewer to listen to the samples, which will be added to our website **iclr-continuous-audio-language-models.github.io** in the next 24 hours.
>
> ---
>
> ## **Answer to the first question**
>
> We have not attempted direct mel-spectrogram generation, but we have added **MELLE** and **Flow-Omni** to the related work.
>
> For speech, we found that **semantic distillation in the VAE is crucial** for strong generation quality. For example, when training on an *acoustic-only* VAE, the WER on LibriSpeech-clean degrades from **1.81% → 2.29%**. Given that spectral representations lack of semanticity, we think that the VAE approach with semantic distillation might be more effective.
>
> ---
>
> ## **Answer to the second question**
>
> Our speech generation models are trained on **~1.8M hours** of in-house data.
> In comparison, our music models are trained on: **~20K hours** (LAION-Disco-12M)  and our text-to-speech model are trained on **88K hours** of public data (details in **Appendix D**)
>
> ---
>
> ## **Answer to the third question**
>
> As requested, we performed a detailed evaluation of our speech compression models (codec and VAE) and reported **PESQ** and **STOI** in **Tab. 1**. The VAE outperforms the 8-RVQ Mimi codec on both metrics.
>
> We will launch a human study shortly and notify the reviewer once results are available.
>
> ---
>
> ### **References**
> [1] Saito et al., SoundReactor: Frame-level Online Video-to-Audio Generation*
> [2] Boffi et al. *How to Build a Consistency Model: Learning Flow Maps via Self-Distillation*

---

### Official Review · Reviewer_mHKG · 2025-11-01

**Soundness:** 2
**Presentation:** 3
**Contribution:** 2
**Rating:** 6
**Confidence:** 3

**Summary:**

This work introduces Continuous Audio Language Models (CALM), a new framework for generating high-quality audio (speech and music). It addresses the primary limitation of current state-of-the-art models, which rely on converting audio into discrete tokens (using RVQ codecs).
CALM bypasses this issue by operating directly in the continuous latent space of a Variational Autoencoder (VAE). Its architecture uses a large Transformer to model the long-term context of the audio. Instead of a slow token-by-token predictor (like an RQ-Transformer) or a multi-step diffusion process, it uses a highly efficient Consistency Model as its head.
Experiments on both speech and music continuation show that CALM achieves higher or comparable quality to strong discrete baselines while being significantly faster and more computationally efficient at inference time.

**Strengths:**

1.Using a  consistency model as the head enables fast single-step generation in a continuous space. Relative to DiTAR-style “continuous latent + AR + diffusion head,” the paper’s contribution is to bring in a consistency-model-style head to reach 1–few steps and to pair this with the long/short context mechanism that specifically stabilizes music-length sequences.

2.Good performance on both speech continuation and music continuation task.

**Weaknesses:**

1.The paper evaluates mainly on continuation. It does not yet demonstrate text-conditioned generation, fine-grained control, or editing, where discrete-token model currently provide a convenient control surface.

2.Asymmetric gains across domains. Music clearly benefits from “noisy long + clean short,” whereas speech shows weaker or no gains from the same trick, suggesting part of the method is tuned to music-style long dependencies rather than being universally optimal.

3.Comparisons are mostly in-house. Many baselines are re-trained under the authors’ own setup (same data, backbone, RVQ depth). This makes the comparison clean but slightly weakens claims about superiority over the latest large-scale discrete audio LMs trained externally.

**Questions:**

Can you include at least one external strong discrete audio LM (not re-trained under your codebase) to demonstrate improvements are not an artifact of your reimplementation?
Since the paper currently focuses on continuation, adding a short section on non-continuation tasks would strengthen the claim that “continuous AR + consistency” is a general recipe.

---

> ### Author Response · Authors · 2025-11-20
> **Response to Reviewer mHKG**
>
> We thank the reviewer for their comments and suggestions. We resubmitted an extended version of the paper with **text-to-speech and text-to-music** experiments that we have run since the submission. These models are compared to external baselines.
> We invite the reviewer to read the updates that are **highlighted in blue**. We address the weaknesses in the following points:
>
> ---
>
> ## Response to the first weakness
>
> Following the reviewer's comment on the fact that we only focus on continuation, we ran experiments in the setups of text-to-music and text-to-speech. However, Classifier Free Guidance (CFG) is detrimental to good quality generation for conditioned generative models. Since, CFG cannot be applied on the trajectory of 1-step consistency models we decide to apply the CFG on the outputs of the Backbone and Short-Context transformers. This method is named **Latent Classifier Free Guidance** and is fully described in the updated version of the paper in **Sec.4.6**. Note that this method has been used as well in the video-to-audio model Soundreactor [1].
>
> ### Text-to-music **(Sec.5.4 & Sec.E.6)**
> - Since our dataset does not have any textual descriptions we use a pretrained CLAP model as a conditioner.
> - We compute the CLAP embedding from the audio at training time.
> - We trained 3 models with CLAP conditioning:
>   - a MusicGen medium
>   - our RQ-Transformer baseline with 32 RVQs
>   - CALM in consistency mode
> - We perform 2 evaluations:
>   1. on our internal test set using CLAP in audio mode (**see Sec.5.4**)
>   2. on the MusicCaps benchmark using textual descriptions at inference time (**see Sec.E.6**) where the model is compared to, MusicLM, MusicGen, AudioLDM2, Noise2Music, Jen-1, MusicFlow.  We notice that even if our goal is not to build the best text-to-music model, **CALM is still competitive** with other models (Tab.11), especially on the FAD and the CLAP score. We think that having at least a few text annotations on our music dataset would benefit our model for text-to-music since the alignment between the text and audio is not perfect in CLAP.
>
> ### Text-to-speech **(See Sec.5.2)**
> For speech, we built a text-to-speech model with the CALM framework (See Sec.5.2) with consistency as well as **Lagrangian Self-Distillation (LSD)** which is a more recent (and more stable) framework of 1-step flow-matching model introduced in [2].
> **Our CALM–LSD model beats baselines (F5TTS [3], DSM [4], DiTAR [5], SALAD [6]) on WER, CER, acoustic quality metrics (human study) and model size.**
>
> However, it suffers from lower speaker similarity, which we believe is caused by our VAE. Indeed, despite having a good reconstruction quality (Tab.1), our VAE has a low measured speaker similarity with reference speech. A human study shows that our model is still preferred in terms of similarity.
>
> **We invite the reviewer to listen to the samples that will be added on our website iclr-continuous-audio-language-models.github.io in the next 24h.**
>
> ---
>
> ## Response to the second weakness
>
> We think that only the music model benefits from the “**noisy long + clean short**” method because unlike the speech model, the music VAE does not have semantic distillation, whereas the speech VAE is trained with a semantic distillation loss from WavLM.
> Past work shows that semantic information guides speech generative models.
> So far, we did not manage to make a music VAE/Codec more semantic and we found that this *“noisy long + clean short”* training significantly improves stability (as heard in the ablations on the sample page).
>
> However, for speech, preliminary experiments with a non-semantic (acoustic only) VAE led to worse results when applying the CALM framework.
> Example: for TTS, WER on Librispeech clean test goes from **1.81% → 2.29%**.
>
> ---
>
> ## Response to the third weakness
>
> We thank the reviewer for acknowledging that our comparison is in a clean setup.
> Now that we have trained text-to-music and text-to-speech models, we have compared our model to baselines.
>
> ---
>
> ## Response to the question
>
> We now have experiments showing that our recipe **“continuous AR + consistency”** applies to conditional tasks such as text-to-speech and text-to-music.
> Comparisons with external models show the competitiveness of our method.
>
> ---
>
> **References**
>
> [1] Saito et al., *SoundReactor: Frame-level Online Video-to-Audio Generation*
> [2] Boffi et al., *How to build a consistency model: Learning flow maps via self-distillation*
> [3] Chen et al., *F5-TTS*
> [4] Zeghidour et al., *Streaming Sequence-to-Sequence Learning with Delayed Streams Modeling*
> [5] Jia et al., *DiTAR*
> [6] Turetzki et al., *Continuous Speech Synthesis using per-token Latent Diffusion*

---

### Author Response · Authors · 2025-11-26
**TTS compute comparison**

We would like again to thank the reviewers for their thorough comments on our paper.

Regarding our text-to-speech model, we have a small update regarding the computational load of our model.
Following the FLOPs computation method from DiTAR (https://arxiv.org/pdf/2502.03930 Sec.A.4) , we obtain a value of **0.5 TFLOPs** for our **CALM TTS** which makes it requiring more than **5 times less compute than DiTAR for similar results**.

We summarize the comparison with the other models to which we compare our method. We take the values as they are reported in the DiTAR paper in Tab.1.

| Model              | TFLOPs |
|--------------------|--------|
| F5 TTS (NFE=32)    | 37.36  |
| DiTAR (NFE=10)     | 2.75   |
| CALM (NFE=1)       | **0.5**    |

We encourage as well the reviewers to have a look at our updated sample page where they can listen to TTS samples at https://iclr-continuous-audio-language-models.github.io

---

### Author Response · Authors · 2025-12-03
**Final Remarks by the authors**

We thank the reviewers for their comments and suggestions that led us to **improving our paper by running new experiments**. As the rebuttal period is ending, we would like to **summarize our contributions**.

Our work is motivated by the idea of getting rid of quantization in the tokenization process of audio for autoregressive models. We started this project by applying the idea of MAR [1] for audio modeling which gave bad results. We came out with a combination of improvements:

1) **Dual-context architecture and noise injection** to improve stability
2) **Head-batch multiplier** to accelerate training, temperature sampling to improve performances
3) **Consistency modeling** instead of diffusion to accelerate inference.

Our method (CALM) achieves **higher quality at lower computational cost than their discrete counterpart**

We originally showcased CALM on the tasks of speech continuation and music continuation:

- For **speech continuation (Tab 2)**, metrics and human studies show CALM generates sentences that are semantically more meaningful than the discrete RQ-Transformer baseline with a better acoustic quality while being 1.3 times faster at inference.
- For **music continuation (Tab 4)**, CALM is 2.2 times faster than a 32-RVQ RQ-Transformer baseline. In addition, it attains a lower FAD score and is rated by human evaluators as producing higher acoustic quality and more appealing music.

## **Improvements coming from the rebuttal period:**

While all reviewers agree that our method leads to good results on these two tasks, **they regret that we did not apply it to the tasks of Text-to-Speech and Text-to-Music** where we can compare ourselves to external baselines (e.g. 3 of them mention the TTS model DiTAR [2]). To answer these concerns, **we applied CALM to these tasks**. In details:

- Because Classifier Free Guidance is detrimental to good quality and prompt adherence for conditional generative tasks, we apply a method that we name **Latent Classifier Free Guidance** that is described in **Sec.4.6**.

- We apply CALM to the **Text-to-Speech** task (we train it on public datasets) and compare it to external baselines which are **DiTAR [2], F5TTS [3], DSM [4], SALAD [5] (see Tab. 3)** on the Librispeech clean test set. While having **fewer parameters** and necessitating **5 times less compute than DiTAR** (0.5 vs 2.75 TFLOPs), **CALM outperforms all baselines** on Word Error Rate, Character Error Rate and Acoustic Quality. Its speaker similarity is slightly worse than the baselines but human raters still judge it more satisfactory than the ground truth. **TTS samples have been added to the webpage.**

- We apply CALM to the **Text-to-Music** task by using a pretrained **CLAP [5]** conditioning in audio mode at training time. At inference time, we can use text to condition the model. In **Tab.6**, we compare our method to a MusicGen model as well as a 32-RVQ RQ-Transformer both retrained with the same CLAP conditioning and show that **CALM is equally good as the 32 RVQ baseline and outperforms MusicGen for a smaller inference cost**. In **Tab11** (page 19) we compare CALM with external models on the **MusicCaps** dataset and show that **our model is competitive with other text-to-music models** that have been trained on text annotated datasets.

**We updated the sample page (https://iclr-continuous-audio-language-models.github.io/) with text-to-speech and text-to-music samples.**

References:

[1] Li et al., Autoregressive Image Generation without Vector Quantization

[2] Jia et al., DiTAR: Diffusion Transformer Autoregressive Modeling for Speech Generation

[3] Chen et al., F5-TTS : A Fairytaler that Fakes Fluent and Faithful Speech with Flow Matching

[4] Zeghidour et al., Streaming Sequence-to-Sequence Learning with Delayed Streams Modeling

[5] Turetzki et al., Continuous Speech Synthesis using per-token Latent Diffusion

[6] Elizalde et al., CLAP: Learning Audio Concepts From Natural Language Supervision

---

### Meta-Review · Area_Chair_grTe · 2026-01-06

**Summary:**

The reviewers in general agree the efficiency advantage of the proposed method.

The concerns are mainly
- Performance is not tested on speech and music tasks
- Novelty on "language modeling on continuous embedding"
- Lack of explaination on some key component like short-context transformer and noise injection

In my opinion, the rebuttal solved the most important concern (lack of experiments and comparison). Regarding the novelty, I believe enhancing the efficiency by putting multiple existing components together is novel for speech related model as real-time is usually a need.

**Reviewer Concerns:**

- Performance is not tested on speech and music tasks

Partially addressed, the authors added experiments and compare with some other models; but the speaker similarity is still lower than baseline. Authors say it is the issue of VAE.

- Novelty on "language modeling on continuous embedding"

Partially addressed

- Lack of explaination on some key component like short-context transformer and noise injection

Partially addressed

In general, the rebuttle added new experiments, which addressed the most important concerns from the reviewer.

**Reviewer Scores:**

mHKG: 6 → 7 (7 means a score between 6 and 8)
because TTS and Music genenration and comparison with other model are added

6NWz: 6 → 7
because conditional generation results are added

vAFX: 6 → 7
because comparison with other model are added

ZVTy: 4 → 6
because comparison with other model are added

---

### Decision · Program_Chairs · 2026-01-26

Accept (Poster)